# Cryo-EM structure of the essential ribosome assembly AAA-ATPase Rix7

Yu-Hua Lo[1], Mack Sobhany[1], Allen L. Hsu[2], Brittany L. Ford [2], Juno M. Krahn[2], Mario J. Borgnia[2] & Robin E. Stanley [1]

Rix7 is an essential type II AAA-ATPase required for the formation of the large ribosomal subunit. Rix7 has been proposed to utilize the power of ATP hydrolysis to drive the removal of assembly factors from pre-60S particles, but the mechanism of release is unknown. Rix7's mammalian homolog, NVL2 has been linked to cancer and mental illness disorders, highlighting the need to understand the molecular mechanisms of this essential machine. Here we report the cryo-EM reconstruction of the tandem AAA domains of Rix7 which form an asymmetric stacked homohexameric ring. We trapped Rix7 with a polypeptide in the central channel, revealing Rix7's role as a molecular unfoldase. The structure establishes that type II AAA-ATPases lacking the aromatic-hydrophobic motif within the first AAA domain can engage a substrate throughout the entire central channel. The structure also reveals that Rix7 contains unique post-α7 insertions within both AAA domains important for Rix7 function.

[1] Signal Transduction Laboratory, National Institute of Environmental Health Sciences, National Institutes of Health, Department of Health and Human Services, 111T. W. Alexander Drive, Research Triangle Park, NC 27709, USA. [2] Genome Integrity and Structural Biology Laboratory, National Institute of Environmental Health Sciences, National Institutes of Health, Department of Health and Human Services, 111T. W. Alexander Drive, Research Triangle Park, NC 27709, USA. Correspondence and requests for materials should be addressed to R.E.S. (email: robin.stanley@nih.gov)

Eukaryotic ribosome assembly is a complex pathway that is dependent upon the aid of hundreds of assembly factors such as ATPases, GTPases, helicases, nucleases, chaperones, and transporters[1–5]. Three AAA (ATPases associated with various cellular activities)-ATPases are required for the assembly of the large ribosomal subunit including Rea1, Rix7, and Drg1[6]. Their proposed function is to utilize the power of ATP hydrolysis to drive the release of specific ribosome assembly factors from pre-60S particles during the 60S assembly pathway[6]. Rix7, the earliest acting ATPase, is proposed to drive release of the essential ribosome assembly factor Nsa1 from nucleolar pre-60S particles[7]. Drg1, the latest acting ATPase, is required for the removal of Rlp24 from cytoplasmic pre-60S particles[8]. Rea1 is involved in several pre-ribosome intermediates, and is required for release of part of the Nop7 complex from early pre-60S particles and release of Rsa4 from nucleoplasmic pre-60S particles[6,9,10]. Structural snapshots of Rea1 have provided invaluable insight into its roles as a checkpoint during processing of the large ribosomal subunit[11]. Meanwhile, a lack of structural information on Drg1 and Rix7 has hindered our understanding of how these motors regulate ribosome assembly.

*Saccharomyces cerevisiae* (*S. cerevisiae*) *RIX7* is an essential gene required for maturation of the large ribosomal subunit[12]. Truncation of the first 14 residues of the N-terminal domain (NTD) of Rix7 leads to the accumulation of ribosome assembly factor Nsa1 on polysomes suggesting a link between Rix7 and release of Nsa1 from pre-60S particles[7]. The mammalian homolog of Rix7 is known as NVL2 (nuclear VCP Like). *NVL*, the gene which encodes for NVL2, has been identified as a prognostic outlier gene for high-risk prostate cancer and shown to confer a risk for mental illness disorders[13,14], signifying a need to understand NVL2's molecular function. NVL2 is a nucleolar protein that associates with pre-60S particles[15]. Overexpression of an ATP-hydrolysis deficient mutant leads to defects in maturation of the large ribosomal subunit[16]. NVL2 has been shown to interact with WDR74[17,18], the mammalian homolog of Nsa1, as well as a number of additional binding partners including the RNA helicase Mtr4[19], nucleolar protein Nucleolin[20], ribosomal protein RPL5[19], and ribonucleoprotein polymerase tERT[21]. Recent studies revealed that WDR74 and NVL2 function together in an early nucleolar pre-rRNA processing step[22]. However it remains unclear if WDR74 is a direct target for remodeling by NVL2 or simply a co-factor.

One of the most important questions about Rix7's molecular function is how does Rix7 pull on substrates to drive their removal from pre-60S particles. Rix7 is a type II AAA-ATPase, composed of a unique NTD followed by tandem well conserved AAA modules known as the D1 and D2 domains respectively. The D1 and D2 domains of Rix7 share homology with other members of the type II AAA-ATPase family, including p97 (Cdc48 in yeast, VAT in archaea), NSF, the Pex1/Pex6 complex, and the ClpB/Hsp100 chaperone family. A number of two-cassette AAA-ATPase structures have been solved, which revealed a common architecture of stacked hexameric AAA rings, while the position of the distinct NTDs varies and can be dependent upon nucleotide and/or substrate binding[23–28]. Despite sharing a similar architecture within the D1 and D2 AAA modules these enzymes can utilize different mechanisms to couple ATP hydrolysis with substrate remodeling[23,26,29,30]. For example, NSF facilitates remodeling of its substrates through conformational changes of its NTD that are driven by nucleotide binding and hydrolysis coupled with partial substrate threading of the D1 domain[23,31]. In contrast recent cryo-EM and in vitro reconstitution studies suggest that Cdc48, Pex1/Pex6, VAT, and ClpB/Hsp100 remodel substrates through a processive threading

mechanism, whereby substrates are pulled/unfolded through the central pore formed by the D1 and D2 domains[29,30,32–34].

Here we report the cryo-EM reconstruction of the Rix7 homohexamer at 4.5 Å resolution. The structure reveals an asymmetric configuration of both the D1 and D2 AAA domains. Through use of a Rix7 variant unable to hydrolyze ATP (Walker B motif in both the D1 and D2 domains), we captured Rix7 in the pre-ATP hydrolysis state with an unexpected polypeptide fragment engaged in the central channel. Five of the six Rix7 protomers grip the polypeptide suggesting a hand-over-hand mechanism for substrate unfolding, providing the first insight into the function of Rix7 as a molecular unfoldase. Despite the absence of the aromatic-hydrophobic motif found in pore loops of unfoldases within the D1 domain, the Rix7 pore loops from both the D1 and D2 domains grip the polypeptide through distinct motifs which are essential for Rix7 function in vivo. The structure also unveils that both the D1 and D2 domains from Rix7 contain structurally distinct post α7 insertions important for Rix7 function.

## Results

**ATPase activity of Rix7 is required for pre-60S biogenesis.** *S. cerevisiae* Rix7 is composed of 837 residues including the NTD, and the D1 and D2 AAA modules (Fig. 1a). Each AAA module in Rix7 contains an α/β subdomain and an α-helical lid subdomain[35]. Rix7 belongs to the classical AAA clade as defined by the addition of a small α-helix (α′) between β2 and α2 (Fig. 1a)[6,35]. The AAA domains of Rix7 include Walker A, Walker B, sensor 1 and arginine finger motifs important for ATP binding and hydrolysis but lack a canonical sensor 2 motif (Supplementary Fig. 1). Rix7 also contains a poorly conserved insertion following α7 in both the D1 and D2 domains, although the significance of these insertions is unknown (Fig. 1a and Supplementary Fig. 1)[6].

To determine the roles of conserved residues within each AAA module for yeast viability and ribosome production we performed genetic complementation assays using a *S. cerevisiae* strain that encodes the tetracycline-inducible promoter (tetO₇) upstream of endogenous *RIX7*. Using this strain expression of endogenous Rix7 can be suppressed by addition of doxycycline (DOX). Yeast which express WT Rix7 (including Flag and Strep tags for detection/purification purposes) from the ARS1-CEN4 YCplac vector[36] grow well in the presence of DOX at 25 °C; however, the empty YCplac vector could not restore growth in the presence of DOX (Fig. 1b). We next tested the complementation of eight individual Rix7 variants including the Walker A, Walker B, sensor 1, and arginine finger motifs in both the D1 and D2 domains. Our data indicate that disruption of the Walker A motif (D1-K252A, D2-K580A) in both the D1 and D2 domains causes a severe growth defect (Fig. 1b). This is in contrast to the Walker B motif (D1-E306Q, D2-E634Q), which causes a mild defect in the D1 domain but a severe defect in the D2 domain (Fig. 1b). These results are in good agreement with a previous study which observed that the Walker B motif in the D1 domain was not essential for viability in *S. cerevisiae*[37]. Disruption of the sensor 1 (D1-N354A, D2-N677A) and arginine finger (D1-R365A, D2-R688A) motifs in both the D1 and D2 domains also caused a severe growth defect (Fig. 1b). To confirm that the variants of Rix7 express in *S. cerevisiae* we analyzed whole cell lysates by western blot and could detect expression of all the Rix7 variants with an anti-Flag antibody (Supplementary Fig. 2a, Supplementary Fig. 6). Collectively these results suggest that ATP binding is required in vivo in both the D1 and D2 domains, while the D2 domain is the major driver of ATP hydrolysis in vivo[37].

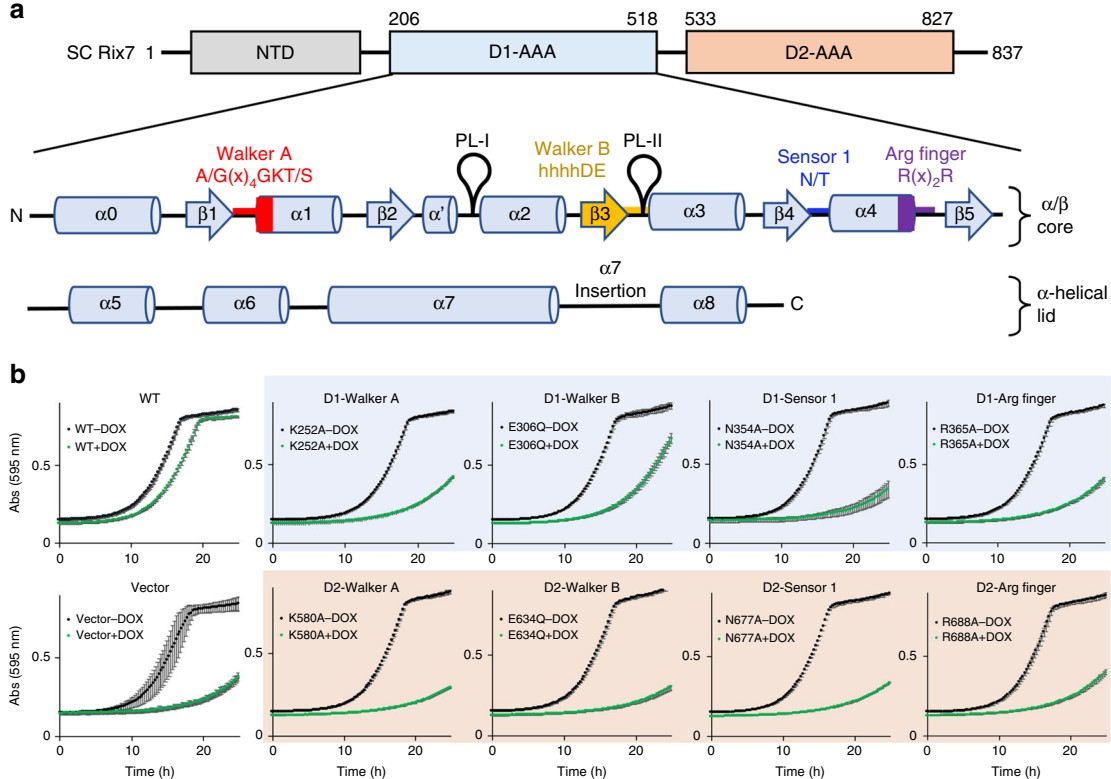

**Fig. 1** Rix7 is an essential type II AAA-ATPase. **a** Cartoon schematic of the three domains of *S. cerevisiae* (SC) Rix7 including the N-terminal domain (NTD, gray), the D1 AAA domain (light blue), and the D2 domain (light orange). Shown in the inset is the secondary structure composition of the D1 AAA domain. Each AAA domain contains an α/β subdomain including the Walker A (red), Walker B (yellow), sensor 1 (blue) and arginine finger (purple) motifs and an all α-helical lid subdomain (α5–α8). The pore loop 1 and pore loop 2 are indicated as PL-I and PL-II, respectively. This figure was adapted from Hanzelmann and Schindelin, 2016[25]. **b** Growth curves of *S. cerevisiae* tetO$_7$–*RIX7* strains transformed with plasmids encoding wild type Rix7, mutants of Rix7, and the ARS1-CEN4 YCplac vector with no insert. Strains were grown in the absence (black) or presence (green) of doxycycline at 25 °C and the absorbance was recorded at 595 nm over a 25 h time period. Each curve is the average of three independent replicates and the error bars mark the standard deviation between replicates. Source data are provided as a source data file

To examine if ATPase activity of Rix7 in the D1 and D2 domains affects ribosome synthesis in *S. cerevisiae* we analyzed sucrose gradients of cytosolic extracts prepared from *tetO$_7$–RIX7* yeast strains expressing Rix7 variants (Fig. 2). We observed a decrease in the 60S peak and appearance of halfmers for the D1 Walker A variant which indicate a defect in mature 60S formation (Fig. 2). Halfmers are shoulders on 80S and polysome peaks that represent 40S ribosomal subunits engaged on mRNA that lack 60S ribosomal subunits. We also observed a decrease in the 60S subunits and the appearance of halfmers for both the D2 Walker A and Walker B variants, whereas free 40S subunits were not affected (Fig. 2). While the D1 Walker B variant does not cause a lethal phenotype we still observed defects in ribosome assembly from the ribosome profile, suggesting that while non-essential, ATP hydrolysis by the D1 domain still plays an important role for Rix7 function in vivo. These results, together with the yeast growth assays (Fig. 1b), reveal that the ATPase activity of Rix7 is required for assembly of the large ribosomal subunit.

**Asymmetric hexameric architecture of Rix7.** Aside from an NMR structure of the very N-terminus of its mammalian homolog NVL2(PDB ID 2RRE)[20] and a deposited structure of the NVL2 D2 AAA domain monomer (PDB ID 2X8A) there was no structural information for the Rix7 D1 domain or the assembled homohexamer. To determine the full-length structure, we utilized single particle cryo-EM to obtain an electron scattering map of a *Chaetomium thermophilum* Rix7 double Walker B mutant (*CT* Rix7$^{E303Q/E602Q}$) and derived a structural model of the hexamer. The EM data were classified and refined without imposing symmetry which resulted in one dominant conformation at 4.5 Å resolution, that contained well resolved density for all 12 AAA domains (Fig. 3a; Supplementary Fig. 3 and Fig. 4). The electron scattering map revealed that Rix7 adopts an asymmetrical stacked hexameric ring configuration composed of six subunits of Rix7 (referred to as protomers P1-P6).

The cryo-EM reconstruction lacks density for the entire NTD of Rix7 (amino acids 1-192), suggesting that this domain is flexible and dynamic. Aside from a helical bundle at the very N-terminus the rest of the Rix7 NTD includes a disordered linker sequence and a nuclear localization sequence[6]. Disorder prediction with IUPred[38] suggests that the entire NTD of Rix7 has an average disorder tendency score of 0.69 (scores above 0.5 indicate disorder) and is predicted to be intrinsically disordered. This is in contrast to VAT/Cdc48/p97 homologs which contain well-ordered NTD regulatory domains[24,25,28] but draws parallels with the ClpB/Hsp100 family in which the NTD appears more flexible and dynamic[32,33,39]. A recent structure of Cdc48 in complex with its cofactor the Ufd1-Npl4 heterodimer suggests that both ATP and cofactor binding move the NTD into the up conformation[40]. We therefore hypothesize that Rix7 specific cofactors are likely required for regulating the intrinsically unstable region of the NTD.

The six protomers within the D1 and D2 domains are arranged in a spiral staircase like pattern (Fig. 3b). The P1 protomer presents a more extended arrangement of the D1 and D2

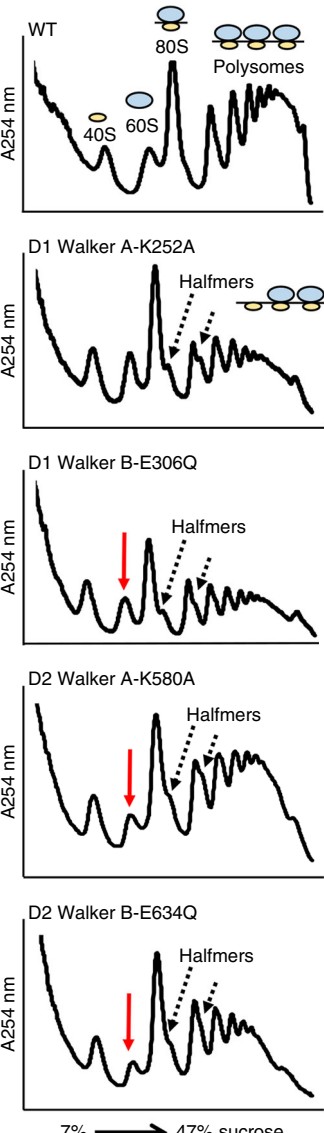

**Fig. 2** ATP hydrolysis by Rix7 is required for ribosome assembly in *S. cerevisiae*. Sucrose gradient polysome profiles of select *tetO7-RIX7* strains grown in the presence of doxycycline. Position of peaks for the 40S subunit, 60S subunit, 80S monosome, and polysomes are indicated on the top profile as a cartoon. The red arrow indicates a decrease in 60S subunits and the dashed lines indicate the presence of halfmers (engaged 40S subunits that lack a 60S subunit as shown in cartoon)

with the D2 protomers but the α-helical lid subdomain of mouse NVL2 is in a different position (Supplementary Fig. 5c).

**Threading the central pore of Rix7**. During the fitting of the model we unexpectedly observed clear density for a polypeptide threading the central pore of Rix7 (Figs. 3c, 4a). There is sufficient density within the pore to accommodate a ~20 residue unfolded peptide spanning the entire 80 Å length of the pore across the D1 and D2 domains. There is no density for the polypeptide above the D1 domain but there is weak density past the D2 domain which may represent accumulation of the polypeptide after it passes through the channel (Fig. 4a). This finding was a surprise because we did not add a substrate to Rix7 prior to preparing cryo-EM grids. The density for the polypeptide is ambiguous and could be part of Rix7 unfolding itself as was observed in the recent structure of VAT[34], or it could be a mixture of non-specific substrates that accumulated during protein expression in *E. coli*. Analysis of our recombinant Rix7 by mass spectrometry did not reveal the presence of any peptides in high abundance other than those arising from Rix7, with the very C-terminus of Rix7 being well represented. While the polypeptide trapped by the Rix7 double Walker B mutant does not represent a real substrate of Rix7, the presence of the polypeptide provides insight into the mechanism of substrate threading and processing by Rix7. Several recent cryo-EM structures of other ATPases that adopt asymmetric spiral ring configurations also contain ambiguous substrate density in their central pores, suggesting that substrates stabilize the asymmetric ring structure[32–34,41].

Five of the six protomers (P1–P5) within both the D1 and D2 domains grip the polypeptide through their pore loops, while P6 does not contact the peptide. The pore loops are arranged in a clockwise spiral configuration around the polypeptide (Fig. 4a, b), likely to maximize the contact area between the AAA motor domains and the substrate[34]. There are two pore loops, pore loop 1 (PL-I) and pore loop 2 (PL-II), that line the central channel within each AAA domain of Rix7 (Figs. 3c and 4a). Protein unfoldases are defined by the presence of a conserved aromatic-hydrophobic-glycine (most often Y/F-V-G) motif within the PL-I of the AAA domain[42,43]. A distinguishing feature of Rix7 and p97 is that both lack the signature aromatic-hydrophobic-glycine motif within PL-I of the D1 domain but these conserved residues are found within PL-I of the D2 domain (Fig. 4d). This has led to a debate about whether or not the D1 domain of p97 is capable of threading a substrate all the way through its central pore[44,45]. The Rix7 cryo-EM structure reveals that even in the absence of the aromatic-hydrophobic-glycine motif, PL-I within the D1 domain of Rix7 (h-S-G, where h is a hydrophobic residue) grips the polypeptide. PL-I within the D2 domain makes extensive contacts with the polypeptide and the K-Y-V-G motif is well ordered in protomers P1-P5 (Fig. 4c). In addition to PL-I, five of the six PL-IIs in the D1 and D2 domain also grip the substrate, albeit with a smaller footprint (Figs. 3c and 4a).

The cryo-EM reconstruction revealed that the arrangement of the pore loops is different in the D1 and D2 domains (Fig. 4b). The separation of PL-I from the highest to lowest protomer in the D1 domain is ~15 Å, whereas PL-I within the D2 domain spans a longer track of ~28Å. The PL-I in the D2 domain are separated from one another by ~6 Å along the channel (Fig. 4b, c), making contact with approximately every second amino acid of the polypeptide. This pore-loop spacing is observed in related AAA rings and represents a conserved feature of translocases[46–48]. Recent studies of the double ring unfoldase Hsp100 also revealed a similar asymmetric architecture between the D1 and D2 domain and biochemical studies of this enzyme suggest that the D2 domain is the main ATP hydrolysis motor[32].

domains compared to P6 protomer, which is more compact (Fig. 3b). The individual α/β subdomains superimpose well with one another but the orientations between the α/β subdomain and α-helical lid subdomains varies between the protomers. Protomers P2, P3, and P4 are the most similar to one another with an RMSD range of 0.71 to 1.22 Å (Supplementary Fig. 5a), while the other protomers have an RMSD range from 1.89 to 6.28 Å over the D1 and D2 domains (Supplementary Fig. 5b). The P6 protomer is the most distinct of the six protomers because it has the most compact orientation of the D1 and D2 AAA domains and the largest angle between the α/β subdomain and α-helical lid subdomains within the D1 domain (Supplementary Fig. 5b). We also compared the individual Rix7 protomers to the deposited coordinates for the mouse NVL2 D2 domain. Overall the individual α/β subdomain of mouse NVL2 superimposes well

**Rix7 pore loops are essential in *S. cerevisiae*.** Based upon the cryo-EM reconstruction, we hypothesized that the pore loops of Rix7 play an important role in substrate translocation in vivo. To determine if the PL-I residues within the D1 and D2 domains of Rix7 are essential in vivo we tested the complementation of a series of Rix7 PL-I variants. First, we replaced the D1 PL-I with the D2 PL-I motif (G278K/M279Y/S280V) and found that this caused a severe growth defect in *S. cerevisiae* (Fig. 4e). Then we made individual point mutants (M279Y, S280Y, S280A) and observed that substitution of an aromatic residue at position 279 is viable. Substitution of an aromatic residue at position 280 is lethal, while substitution with an alanine causes a moderate growth defect (Fig. 4e). The high degree of conservation across Rix7 homologs of this serine residue in PL-I that is not present in related AAA-ATPases (Fig. 4d and Supplementary Fig. 1) suggests it has an important role specific for Rix7 function. Next, we targeted residues within the D2 PL-I, containing the conserved aromatic-hydrophobic motif. Individual alanine mutations of the aromatic (Y607A) or hydrophobic (V608A) residues results in a lethal phenotype in yeast. However, alanine mutation of the well conserved lysine (K606) causes a moderate growth defect in *S. cerevisiae* (Fig. 4e). Expression of the Rix7 pore loops mutants in *S. cerevisiae* was confirmed by western blot analysis of whole cell lysates (Supplementary Fig. 2b, 2c, and 6). Taken together our results indicate that the pore loops within Rix7 are distinct from one another to support the different functions of the D1 and D2 domains. The sequence and structural differences between the D1 and D2 domains of Rix7 further support the role of the D2 domain as the main motor which drives translocation of the unfolding substrate.

**The seam protomer and the nucleotide-bound state.** The cryo-EM reconstruction of Rix7 also revealed a seam subunit, which has been observed in other recent substrate bound type II AAA-ATPases[32–34]. We defined the P6 protomer as the seam subunit because it does not contact the polypeptide or closely pack with the neighboring AAA domain thereby creating a seam or break in the hexameric ring[34]. The P6 seam protomer within both D1 and D2 domains of Rix7 breaks the helical arrangement and detaches from the bound polypeptide (Fig. 4a, b). The cryo-EM map in the area of the P6 protomer has less defined density and a lower local resolution than the other five protomers of Rix7 suggesting that this subunit is more mobile (Supplementary Fig. 4c). The P6 protomer, in the D1 domain, is partially detached from P5 and does not tightly pack against P1, thus creating a seam between the P6 and P1 protomers (Fig. 5a). In the D2 domain the P6 protomer closely packs with P1 but not P5, thus creating a seam between the P5 and P6 protomers in the D2 rings (Fig. 5b).

During the purification of recombinant Rix7 and the preparation of cryo-EM grids, we did not add ATP or a non-hydrolysable ATP variant. To determine if nucleotide was present in each of the 12 ATP binding sites within our cryo-EM reconstruction, we prepared a difference map between the nucleotide free hexamer model and the cryo-EM map. Clear difference density attributed to nucleotide is present in 10 of the 12 ATP binding sites (Fig. 5a, b), suggesting that nucleotide binding is important for polypeptide engagement in the central channel. In the D1 AAA domain, ATP is present in the nucleotide pockets of all six protomers, whereas only four nucleotides were observed in the nucleotide binding pockets of P1-P4 from the D2 AAA domain. There is no density for ATP in the nucleotide binding pocket of P5 from the D2 domain. The Walker A motif (also known as the P-Loop) adopts a different orientation from the other AAA domains indicating that this AAA domain is in the apo state (Fig. 5c, d). The α7 helix from P5 partially blocks the nucleotide binding

pocket in the D2 domain (Supplementary Fig. 5c), suggesting a state following ATP hydrolysis and release of ADP. Superposition of the phosphate bound mouse NVL2 D2 domain structure (PDB ID 2X8A) onto the P4 protomer D2 domain also unveils a clash between the α7 helix and the ATP binding pocket suggesting that the α7 helix must move to accommodate nucleotide binding (Supplementary Fig. 5c). There is less defined density at the nucleotide binding pocket in the D2 domain of the P6 seam protomer. The P6 protomer has a lower local resolution and poor density so we cannot distinguish between a partial or fully occupied ATP-bound state (Fig. 5b).

**Insertions following α7 in the D1 and D2 domains.** Aside from distinct NTD domains, another feature that distinguishes Rix7 from other type II ATPases is the presence of an insertion following helix α7 in both the D1 and D2 domains. In contrast to Rix7, p97 only contains an α7 insertion in the D2 AAA domain (Supplementary Fig. 1). Previous studies have suggested the α7 insertion in the D2 domain of p97 is important for ATP hydrolysis and substrate recognition[25] but the significance of these insertions in Rix7 is currently unknown. Within the Rix7 substrate-engaged structure the two α7 insertions are structurally different from one another (Fig. 6a, d). Helix α7 in the D1 domain is extended and bends back towards the neighboring AAA protomer (Fig. 6a). There is also the addition of a small α-helix (α7′) that packs against α0 from the neighboring AAA protomer. The α7 insertion in the D1 domain would sterically clash with the position of the NTD of p97 in the ADP-bound state (Fig. 6b). While we do not observe any density for the NTD domain of Rix7, the location of the post α7 insertion in the D1 domain suggests that the NTD of Rix7 occupies a location distinct from the NTD of p97. The additional interface with the neighboring AAA module also suggests a putative role for the post α7 extension in stabilizing the hexamer and one could imagine straightening of the bent helix would break apart this interface (Fig. 6a). In contrast to the D1 domain, the post α7 insertion within the D2 domain does not bend back and make as extensive contacts with the neighboring AAA module (Fig. 6d) and aligns well with the post α7 insertion in p97 (Fig. 6e). The difference in the post α7 insertions contributes to the difference in buried surface area between the protomers in the D1 and D2 domains. With the exception of the seam protomer, the D1 protomers have an average buried surface area of 1700 Å$^2$ while the protomers in the D2 domain have an average buried surface area of 1300 Å$^2$.

The cryo-EM reconstruction suggests that the post α7 insertions in the D1 and D2 domains of Rix7 play an important role in Rix7 function. To determine if the post α7 insertions in the D1 and D2 domains of Rix7 are essential in vivo we tested the complementation of post α7 insertion deletions. In the D1 domain, we deleted the residues predicted to form the additional α7′ helix from *S. cerevisiae* (residues 482–492; Supplementary Fig. 1). Deletion of this helix causes a moderate growth defect indicating that it is important for optimal Rix7 function (Fig. 6c). In the D2 domain, we deleted residues 771–784 from *S. cerevisiae* (Supplementary Fig. 1) and found that this post α7 deletion causes a severe growth defect in *S. cerevisiae* confirming that this extension region is required for Rix7 function (Fig. 6f). This work establishes that the Rix7 post α7 insertions play important roles in Rix7 function in vivo.

## Discussion

Rix7 is a poorly characterized type II AAA-ATPase that plays an essential role in the assembly of the large ribosomal subunit. While related to other type II AAA-ATPases, how Rix7 utilizes its molecular AAA motors to extract proteins from pre-60S particles has

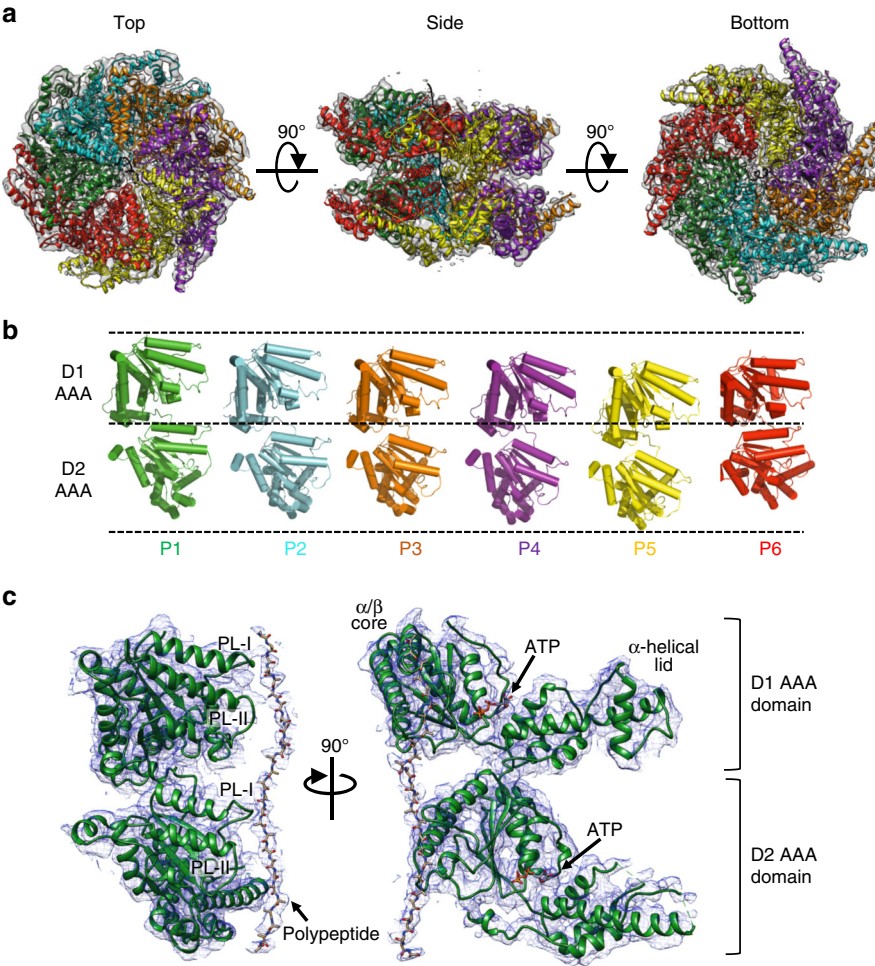

**Fig. 3** Rix7 forms an asymmetric closed ring double hexamer. **a** Top, side, and bottom views of the final Rix7 cryo-EM reconstruction shown in gray. The pseudo atomic model of Rix7 is shown overlaid with the reconstruction. Individual AAA protomers are colored as follows P1 (green), P2 (cyan), P3 (orange), P4 (purple), P5 (yellow), and P6/Seam (red). **b** View of the individual protomers from the hexamer. The view from each protomer was generated by successive 60° rotations along the y axis. Dashed lines are defined as the highest position of D1 domain (P1 protomer), the linker position between D1 and D2 (P1 protomer), and the lowest position of D2 domain (P5 protomer). **c** The pseudo atomic model of the Rix7 P1 protomer shown as a cartoon with the cryo-EM reconstruction overlaid. This illustrates the fit of the individual protomer within the cryo-EM map and the quality of the reconstruction. Also shown is the additional density observed for the polypeptide in the pore. The pore loop 1 and pore loop 2 are indicated as PL-I and PL-II, respectively in both the D1 and D2 domains. The position of ATP in the nucleotide binding pocket is indicated

remained unknown. Through use of a double Walker B mutant we trapped Rix7 with a polypeptide threading the entire central channel formed by the D1 and D2 domains. While the identity of this polypeptide is unclear, the peptide mimics the presence of a substrate in the central channel. In addition to the D1 and D2 AAA domains, type II AAA-ATPases typically contain N-terminal domains important for substrate recruitment[26,29,32,49]. We were unable to observe any density for the six NTDs of Rix7 in our 3D cryo-EM reconstruction, which was not surprising given the high degree of predicted disorder in this domain. We previously identified that Rix7's putative substrate Nsa1 (WDR74 in mammals), associates with the D1 AAA domain of Rix7 (NVL2 in mammals) and not the NTD, suggesting that Rix7 engages substrates in a different manner than other type II AAA-ATPases[18]. While the NTD of Rix7 is dispensable for Nsa1 binding, it is essential for Rix7 function, Nsa1 release from pre-60S particles, and nuclear/nucleolar localization[6,7,20]. The molecular role of the NTD remains undefined but we speculate that it functions as a molecular gate to allow access for the substrate to bind the D1 domain and thread through the central pore.

Our cryo-EM reconstruction establishes that Rix7 is a molecular unfoldase that utilizes conserved pore loops lining the central channel to pull on substrates. The central pore loop residues of Rix7 adopt a clockwise staircase-like arrangement, in which five of the six protomers grip the substrate in the channel while one protomer, the P6/Seam, does not contact the substrate (Fig. 7a). The P6 protomer has less well-defined density than the other protomers suggesting that it is flexible because it is mobile and positioned in a "transition state" where it is moving back up to the top of the ring, to grip the substrate, along with the pore loops of the other protomers (Fig. 7b). Meanwhile, ATP hydrolysis from the D2 domain in P5 (lowest position protomer) powers the movement of the substrate and then following nucleotide and substrate release it takes the position of the P6 protomer (Fig. 7c). Alternating grips between power strokes will continue in a clockwise rotary style fashion to drive the translocation of the substrate through the central pore (Fig. 7c). Similar processive substrate translocation models have been proposed following a number of recent cryo-EM structures of single and double ring AAA-ATPases[32–34,41,50,51].

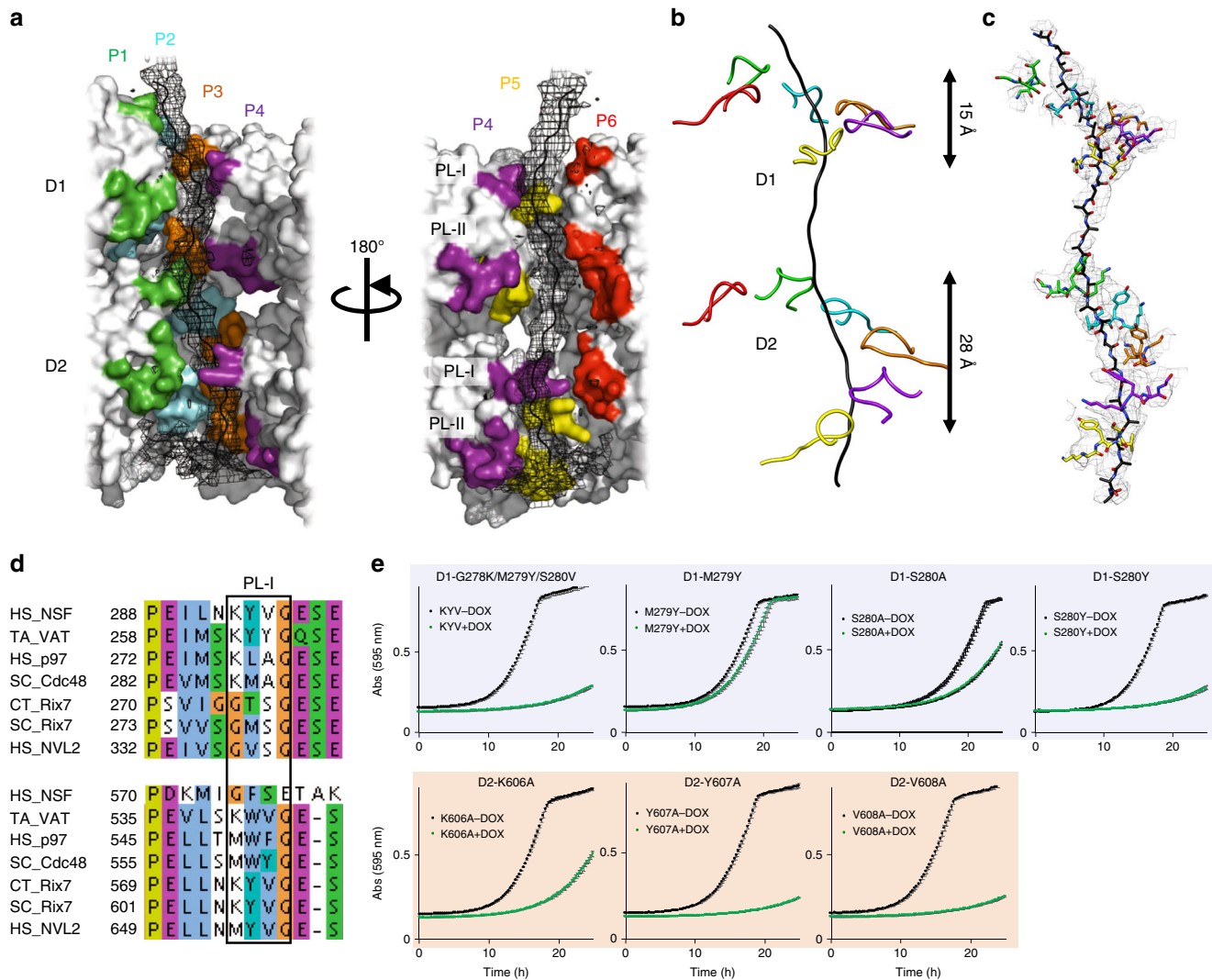

**Fig. 4** Rix7 pore loops are essential for function in vivo. **a** Rix7 engages a substrate in its central pore. Difference map generated from a substrate free model of Rix7 is shown in black mesh, and the modeled peptide is shown as a black cartoon. The pseudo-atomic model of Rix7 is shown as a surface representation in gray, with the pore loop residues colored by the individual protomers. The pore loop 1 and pore loop 2 are indicated as PL-I and PL-II, respectively. **b** Simplified view of PL-I engaging the substrate in the D1 and D2 domains. The D2 PL-I engages more of the polypeptide than the D1 PL-I. **c** Conserved motifs from D1 and D2 PL1 (shown as sticks) grip the polypeptide (shown as sticks). The cryo-EM map is overlaid on top. **d** Multiple sequence alignment of the D1 and D2 domains from several type II AAA family members, including NSF, VAT/p97/Cdc48, and Rix7/NVL2. Only the alignment surrounding PL-I (boxed) in each AAA domain is shown. Abbreviations are as follows: *Homo sapiens* (HS), *Chaetomium thermophilum* (CT), *Saccharomyces cerevisiae* (SC) and *Thermoplasma acidophilum* (TA). Alignments were done in Clustal omega[69] and illustrated with JalView[70]. **e** Growth curves of *S. cerevisiae* tetO₇-RIX7 strains transformed with plasmids encoding pore loop mutants of Rix7. Strains were grown in the absence (black) or presence (green) of doxycycline at 25 °C and the absorbance was recorded at 595 nm over a 25 h time period. Each curve is the average of three independent replicates and the error bars mark the standard deviation between replicates. Source data are provided as a source data file

While our structure revealed that Rix7 draws many parallels with other molecular unfoldases our studies also unveiled several unique distinctions between the D1 and D2 AAA domains of Rix7, including the arrangement and sequence of the central pore loops, the requirements for ATP hydrolysis, and the Rix7 specific post α7 helix insertions. The conserved residues in PL-I within Rix7 contribute to the distinction between the D1 and D2 domains. Rix7, p97, and NSF likely evolved from a common ancestor, the archaeal unfoldase VAT[43]. VAT contains the canonical aromatic-hydrophobic residues in both its D1 and D2 domains while, Rix7 and p97 lack the aromatic-hydrophobic motif in the D1 domain and NSF lacks this motif in the D2 domain, suggesting that the pore loops of these motor proteins evolved for different functions. Our data revealed the importance of a well-conserved serine residue from the D1 PL-I signature

motif (G-h-S-G, where h is a hydrophobic residue). Moreover, we found that the aromatic-hydrophobic residues from the D2 PL-I are essential and adopt a more extended spiral arrangement than those within the D1. This leads to a more extensive interface between the D2 pore loops and the substrate and suggests that the D2 domain is the major motor driving translocation of the unfolding substrate through successive rounds of ATP binding/hydrolysis.

The requirements for ATP hydrolysis also differ between the D1 and D2 domains. Through yeast complementation assays we show that the Walker A, arginine finger, and sensor 1 motifs are essential in vivo in both the D1 and D2 domains while the Walker B motif, which is required for ATP hydrolysis is only essential in the D2 domain. In vitro reconstitution experiments with Cdc48 and its substrates have revealed that substrate binding decreases

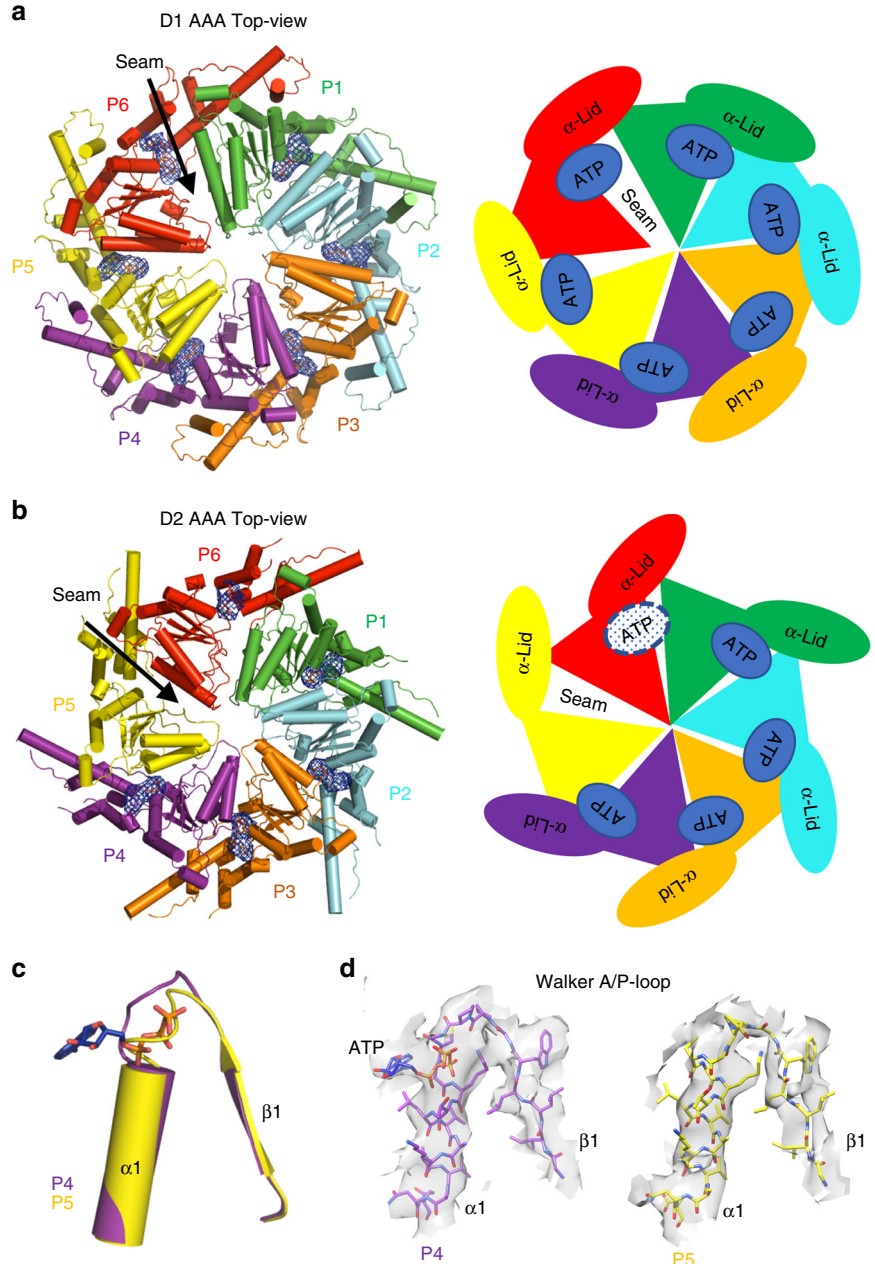

**Fig. 5** Rix7 has an asymmetric seam. **a** Top-view of the protomers of Rix7 in D1 domain colored as indicated and shown as a cartoon (left) and model (right). Shown in mesh is the Fo-Fc difference map at 4 sigma, calculated from the experimental cryo-EM map and a Rix7-nucleotide free model. There are nucleotides present in the nucleotide binding pockets of all six protomers in the D1 domain. **b** Top-view of the protomers of Rix7 in D2 domain shown as a cartoon and model, with the nucleotide shown as in a. ATP is present in protomers P1-P4. There is no nucleotide present in P5, and P6 has less defined density in the nucleotide pocket, which could be a partial occupancy of ATP. **c** The D2 domains from P4-P5 were superimposed on their α/β subdomains and shown is an overlay of the Walker A (P-Loop) motif from P4 and P5, which is located between β1 and α1 in the α/β subdomain. ATP from P4 is shown as a stick model. The Walker A motif from P5 is not in a position that is compatible with ATP binding. **d** Densities for the Walker A (P-Loop) motif in D2 domain from P4 (left) and P5 (right) protomers. The ATP bound in P4 protomer is shown as a stick

ATP hydrolysis in the D1 domain and stimulates ATP hydrolysis in the D2 domain leading to the unfolding/translocation of the substrate through the central pore[29]. Yeast variants of Cdc48 with D1 Walker B mutants are also viable but show a reduced growth rate[52], similar to what we observe with Rix7. Therefore, while ATP hydrolysis within the D1 domain of Rix7 is not essential it still supports Rix7 function in vivo.

The cryo-EM reconstruction of Rix7 also unveiled another distinction between the D1 and D2 domains of Rix7, the post α7

helix insertion in the α-helical lid subdomain. The α-helical lid subdomain of AAA-ATPases is composed of 4 α-helices (α5-α8), however this region of the AAA module can be variable and other ATPases such as p97 and ClpB have insertions in this region[25]. Based upon the cryo-EM reconstruction, Rix7 contains a specific post α7 insertion within the D1 domain that makes additional contact with the neighboring AAA domain, suggesting it could be important for stabilization of the hexameric ring. This is in contrast to the post α7 insertion within the D2 domain which

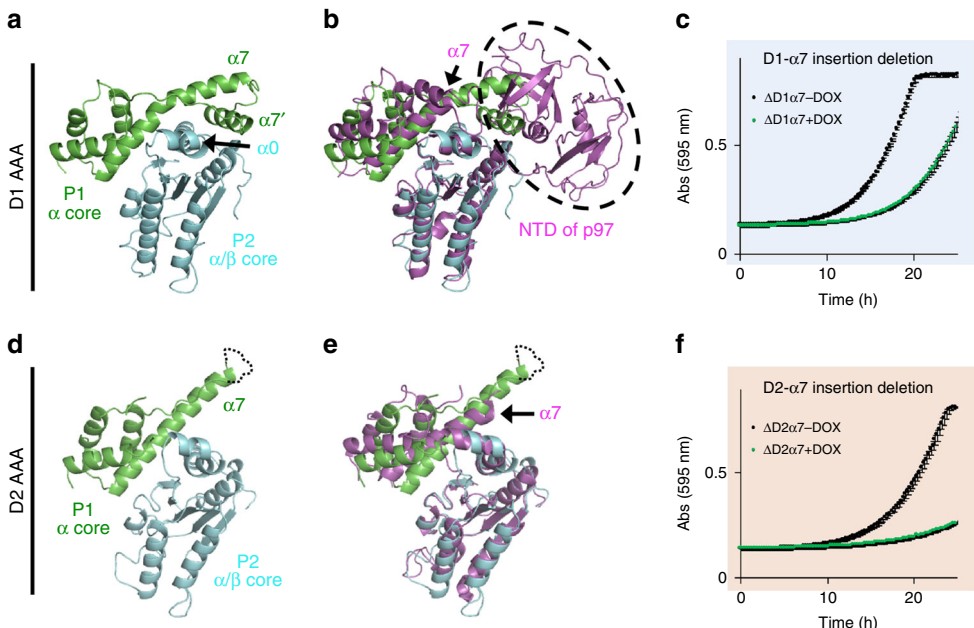

**Fig. 6** Rix7 specific post α7 insertions. **a** Post α7 insertion in the D1 AAA domain of Rix7. Close-up view of the interface between the P1 α-helical subdomain (green) and the P2 α/β subdomain (cyan). Following α7 is an additional helix, α7′ which contacts the neighboring P2 α/β subdomain. **b** The D1 AAA domain of p97 (PDB ID 5FTK, colored in magenta) was superimposed onto the D1 domain of the P2 protomer of Rix7. The location of the NTD of p97 is shown in the dashed oval. **c** Growth curves of *S. cerevisiae tetO₇-RIX7* strains transformed with a plasmid encoding the Rix7 △D1α7 variant. Strain was grown in the absence (black) or presence (green) of doxycycline at 25 °C. Each curve is the average of three independent replicates and the error bars mark the standard deviation between replicates. **d** Post α7 insertion in the D2 AAA domain of Rix7. Close-up view of the interface between the P1 α-helical subdomain (green) and the P2 α/β subdomain (cyan). In contrast to the D1 domain, the D2 α7 insertion does not bend back to form additional contacts with the P2 α/β subdomain. **e** D2 AAA domain of p97 (magenta) was superimposed onto the D2 domain of P2 protomer of Rix7. The D2 α7 helix of p97 is indicated by the arrow. **f** Growth curves of *S. cerevisiae tetO₇-RIX7* strain transformed with a plasmid encoding the Rix7 △D2α7 variant. Strain was grown in the absence (black) or presence (green) of doxycycline. Each curve is the average of three independent replicates and the error bars mark the standard deviation between replicates. Source data are provided as a source data file

does not make as extensive interactions with the neighboring AAA domain. Taken together, our results suggest that the D1 and D2 AAA domains of Rix7 play distinct molecular roles in vivo. The D1 domain is important for hexamer formation, substrate recruitment, and initial threading of the substrate into the channel, while the D2 domain is the driver of ATP hydrolysis and unravels the substrate through successive rounds of ATP hydrolysis (Fig. 7). Rix7 is analogous to a sports car with rear wheel drive. Imagine the D1 domain as the front of the car and the D2 domain as the rear. Just like in the sports car the D1 domain steers Rix7 so that it can guide the substrate into the central channel, while the D2 domain contains the motor and is responsible for powering the movement of the substrate.

In summary, our work establishes Rix7 as a nucleolar molecular unfoldase that utilizes the D1 and D2 AAA domains to thread substrates through its central pore via a processive hand-over-hand mechanism. While our work provides a wealth of information about Rix7 unfoldase activity there are numerous unanswered questions which await further discovery. For example, Rix7 mutants display a dominant phenotype in *S. cerevisiae* even in the absence of Nsa1, suggesting that Rix7 has additional molecular targets[37]. Does Rix7 fully unfold its substrates or simply pull on disordered/unfolded regions to drive removal from pre-60S particles? Our previous work established that Nsa1 contains a flexible C-terminal tail, thus one intriguing possibility is that Rix7 could pull on the tail of Nsa1 to remove it from pre-60S particles[18]. This work lays the foundation for future studies on determining the molecular substrates of Rix7 and the molecular mechanisms that link protein unraveling by Rix7 with assembly of the large ribosomal subunit.

## Methods

**Cloning of Rix7 variants**. *S. cerevisiae* Rix7 along with its endogenous promoter and terminator were cloned into the YCplacIII yeast centromeric plasmid[36]. A list of all primers used in this study is shown in Supplementary Table 1. A 3x N-terminal Flag tag and C-terminal Strep tag were added to Rix7 for downstream detection/purification applications. This plasmid was used as the template for Q5 site directed mutagenesis to prepare all Rix7 mutants (Supplementary Table 2). All plasmids were verified by DNA sequencing (GeneWiz). DNA encoding for the D1/D2 Walker B variant of *C. thermophilum* Rix7 was codon optimized for *E. coli* expression (GenScript) and inserted into the pet24b expression plasmid containing a C-terminal hexa-histidine tag (Supplementary Table 2).

**Yeast growth and proliferation assays**. The *S. cerevisiae* strain encoding a tetracycline-titratable promoter (*tetO₇*) upstream of the endogenous *RIX7* gene was obtained from Open Biosystems (GE Dharmacon). A 5xFlag tag was added onto the C-terminus of endogenous *NSA1* in the *tetO₇-RIX7* strain for downstream detection purposes with the pFA6a-5FLAG-natMX6 vector[53]. Expression of endogenous *RIX7* was suppressed by supplementing the YPD media with doxycycline (DOX, 120μgml⁻¹). The *tetO₇-RIX7_NSA1-5xFlag* strain was transformed with plasmids encoding Rix7 mutants (Supplementary Table 3). The Rix7 mutants were expressed from the ARS1-CEN4 YCplac vector transformed into the *tetO₇-RIX7* strain. We tested the complementation of the Rix7 mutants by suppressing endogenous WT Rix7 expression with DOX following established protocols[54,55]. The growth curves were generated from YPD cultures (100 μl) with and without DOX inoculated at an OD600 of 0.05 and incubated at 25 °C for 25 h. Growth was monitored by measuring the absorbance at 595 nm every 15 min using an Infinite 200 Pro (Tecan). All growth assays were performed in triplicate.

**Western blots**. Transformed yeast cultures (Supplementary Table 3) were grown at 30 °C in the presence of DOX (120 μg ml⁻¹) to an OD600 of 0.2. Cells were then harvested and protein extraction was performed as described in Zhang et al.[56]. Western blots were then performed as described in Romes et al.[57] using either anti-Flag (Sigma-Aldrich, #f425) antibody (1:1000 dilution), or anti-alpha Tubulin Yol1/34 (Abcam, #ab6161) antibody (1:500 dilution) as primary antibodies. Secondary antibodies used for visualization were either goat anti-rat IgG (R&D Systems, #HAF005) or goat anti-rabbit IgG (Sigma-Aldrich, #A0545) conjugated HRP

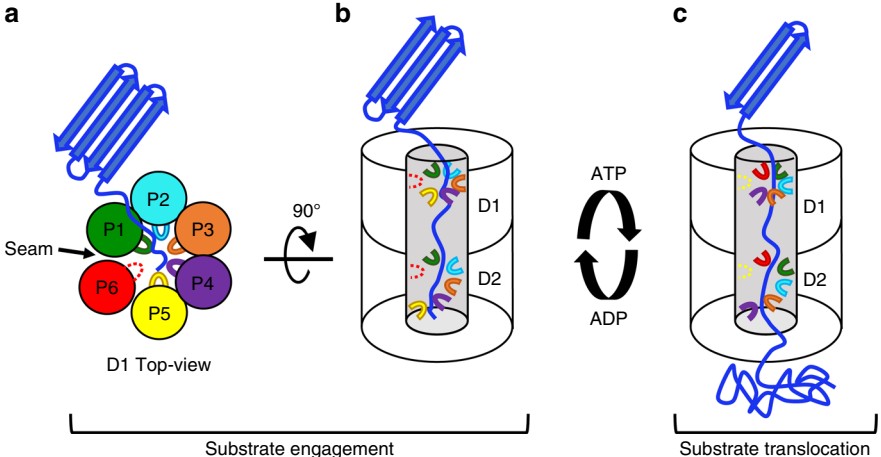

**Fig. 7** Model of processive unfolding by Rix7. **a** In the presence of ATP, a target substrate is engaged by the D1 domain. Five of six pore loops (P1-P5) contact the substrate and thread it into central pore. The P6 protomer (seam subunit) does not closely pack with the neighboring AAA domain thereby creating a seam in the hexameric ring and detaching from the central substrate. **b** With the exception of the seam protomer, the D2 pore loops adopt a more extended clockwise helical arrangement to contact substrate. **c** ATP hydrolysis by the lowest position protomer (P5 protomer, yellow) drives release from the substrate, while the seam protomer re-engages further along the substrate. Pore loops engaging the substrate are shown as a solid line, while pore loops that do not engage the substrate are shown as a dashed line

antibodies at manufacturer recommended dilutions. Protein bands were visualized with ECL Plus Western blotting detection reagent (GE Healthcare).

**Sucrose gradient and polysome profiling.** Starter cultures of transformed *tetO₇-RIX7* (Supplementary Table 3) were grown at 30 °C in the presence of DOX (120 μg ml⁻¹) for 16 h. 1 L cultures of YPD and DOX (120 μg ml⁻¹) were inoculated to an OD of 0.05 and then grown at 30 °C to an OD of ~0.6. Cycloheximide (0.1 mg ml⁻¹) was added to the cultures and the cultures were incubated for 5 min on ice before harvesting cells by centrifugation. Cells were resuspended in extraction buffer (20 mM Tris-HCl pH 7.4, 60 mM KCl, 10 mM MgCl₂, 1 mM DTT, 1% [v/v] Triton X-100, 0.1 mg ml⁻¹ cycloheximide, 0.2 mg ml⁻¹ heparin [ammonium salt]) and then disrupted with glass beads on a Disruptor Genie (Scientific Industries) for 5 min at 4 °C. Lysate was clarified at 8000 × g for 5 min at 4 °C. RNA was quantitated using a Qubit fluorometer (Invitrogen) and 0.1 mg of total RNA was loaded onto a 7–47% sucrose gradient. Gradients were subjected to 260,110 × g for 2.5 h at 4 °C prior to fractionation (Brandel). All sucrose gradient experiments were performed in triplicate and a representative trace is shown in Fig. 2.

**Expression and purification of Rix7.** The *CT* Rix7 expression plasmid was transformed into BL21(DE3)* cells. Cells were grown at 37 °C to mid log phase (OD₆₀₀ = 1.0), induced with 1 mM IPTG and then harvested 3.5 h after induction. Harvested cells were resuspended in Lysis buffer (50 mM Tris-HCl pH 7.5, 500 mM NaCl, 10% glycerol, and 10 mM MgCl₂) and then lysed by sonication. After centrifugation at 26,916 × g for 45 min, the supernatant was loaded onto TALON metal affinity resin (Clontech) equilibrated in Lysis buffer. Rix7 was eluted with Elution buffer (50 mM Tris-HCl, pH 7.5, 500 mM NaCl, 5 mM MgCl₂, 5% glycerol and 200 mM imidazole) after washing with 10 column volumes of Lysis buffer. The eluate was then further purified on a Superdex 200 column (GE Healthcare) equilibrated with Buffer A (20 mM Tris-HCl pH 8, 150 mM NaCl, 5 mM MgCl₂ and 0.5 mM DTT). Column fractions were analyzed by SDS-PAGE; those containing *CT* Rix7 were pooled and concentrated to 10 ml then injected onto a MonoQ column (GE Healthcare). Elution was performed by gradient increase in salt concentration (150–500 mM NaCl). To confirm the presence of hexamer, *CT* Rix7 was further purified using a Superose 6 increase column (GE Healthcare) in Buffer A. Fractions were analyzed by SDS-PAGE and then used for making cryo-EM grids.

**Cryo-EM specimen preparation and data collection.** Cryo-EM specimens were prepared on holey carbon grids (C-flat™ 1.2/1.3-4Cu-50, Protochips) using standard preparation techniques. Following SEC fractionation protein concentration was adjusted to ~0.6 mg ml⁻¹ and a 5 μl aliquot was applied to the carbon side of a freshly glow-discharged grid (30 s, 15 mA at 0.38 mBar in a PELCO Glow Discharge system) in an environmentally controlled chamber (95% RH and 25 °C, Leica EMGP). Excess liquid was blotted for 3 s with filter paper (Whatman #1) from the same side as applied and plunged into liquid ethane (91 K). Specimens were imaged in a Titan Krios transmission electron microscope (Thermo-Fisher) operated at 300 keV and equipped with a Falcon 3EC detector (Thermo-Fisher). Data was collected using EPU at a nominal magnification of ×59,000

corresponding to a physical pixel size at specimen level of 1.39 Å pixel⁻¹. Images were collected under focus in the range of 1.2–2.7 μm. A total of 40 electrons Å⁻¹ were recorded in counting mode at the dose rate of 0.8 electrons pixel⁻¹ s⁻¹, and stored as movies consisting of 45 fractions of equal dose (Table 1).

**Cryo-EM image processing.** The motioncor2[58] frame alignment program was used to perform motion correction and dose-weighting, with the first and last two frames of the movie removed. Frame alignment was performed on 5 × 5 tiled frames. Defocus and astigmatism of the resulting aligned movies were estimated using CTFFIND4[59,60]. For the reconstruction of *CT* Rix7 mutant (E303Q/E602Q), all processing steps, 2D and 3D classification and 3D refinement were performed with RELION[61]. A total of 693,095 putative molecular images were automatically picked in RELION using a Gaussian blob as reference. A first round of reference free 2D classification was used to remove contamination and to separate the particle set into top and side views (287,671 and 56,998 particles respectively). Further rounds of 2D classification were performed separately for top and side views (Supplementary Fig. 3b and Table 1). Selected top and side class averages were used to create an initial reference volume using the program e2initialmodel.py from EMAN2[62]. This map was refined in 3D against the 215,413 molecular images extracted from the selected classes. Two rounds of refinement were performed, first using C2 symmetry and then relaxing the symmetry completely (C1). An additional round of 3D classification was performed to further remove contaminating images and in a fruitless attempt to detect potential conformational variants. A subset of 64,386 molecular images was extracted from a well-behaved class which showed a complete double layer of hexameric rings. The final electron scattering density map, obtained by 3D refinement of the selected particles in Class 1, has a resolution estimate of 4.5 Å based on the 0.143 cutoff for the Fourier Shell Correlation between two half-maps independently refined in RELION (Supplementary Fig. 4a and Table 1). The maps were postprocessed in RELION and are shown after B-factor sharpening. Figures were prepared with UCSF Chimera[63] and pymol[64].

**Model building and refinement.** The Rix7 cryo-EM reconstruction has an estimated resolution of 4.5 Å. At this resolution we observe clear density for all the major features of the 12 AAA domains of Rix7. We also observe some side chain density for well-ordered bulky residues. To build an initial model of Rix7, we took the coordinates from the D1 and D2 domains from a recent p97 structure (PDB ID 5C1A) and manually fit the α/β and α subdomains into each AAA protomer using COOT[65]. The fit was then improved using rigid body refinement. To build a pseudo atomic model of Rix7, we generated a homology model using Swiss Modeler[66]. This model was then superimposed upon the initial model and the fit was improved with rigid body and phased reciprocal space refinement in CNS[67]. Regions of Rix7 that differ from p97 such as the post α7 insertion were built de novo in COOT. Other minor adjustments were made to the model following iterative rounds of manual building and refinement in COOT and CNS. Refinement incorporated restraints for 65% of the residues with p97 reference coordinates (PDB: 5C18, 5C19), NCS restraints among chains for 644 residues in 8 subgroups, and secondary structure restraints for α helices and β sheets. The model includes side chains only for completeness, and are restrained to the reference model where

**Table 1 Cryo-EM data collection, refinement and validation statistics**

| | Rix7 D1/D2 Walker B Mutant (EMD- 9063) (PDB 6MAT) |
|---|---|
| **Data collection and processing** | |
| Magnification | 59,000 |
| Voltage (kV) | 300 |
| Electron exposure (e–/Å²) | ~40 |
| Defocus range (μm) | −1.2 to −2.7 |
| Pixel size (Å) | 1.39 |
| Symmetry imposed | C1 |
| Initial particle images (no.) | 693,095 |
| Final particle images (no.) | 64,386 |
| Map resolution (Å) | 4.5 |
| FSC threshold | 0.143 |
| Map resolution range (Å) | 4–8 |
| **Refinement** | |
| Initial model used (PDB code) | 5C1A |
| Model resolution (Å) | 4.5 |
| FSC threshold | 0.143 |
| Map sharpening $B$ factor (Å²) | −183.6 |
| **Model composition** | |
| Chains | 7 |
| Non-hydrogen atoms | 27,034 |
| Protein residues | 3475 |
| Ligands (ATP) | 11 |
| **$B$ factors (Å²)** | |
| Protein | 97.98 |
| Ligand | 39.34 |
| **R.m.s. deviations** | |
| Bond lengths (Å) | 0.0117 |
| Bond angles (°) | 1.57 |
| **Validation** | |
| MolProbity score | 1.91 |
| Clashscore | 2.81 |
| Poor rotamers (%) | 2.79 |
| **Ramachandran plot** | |
| Favored (%) | 91.56 |
| Allowed (%) | 6.40 |
| Disallowed (%) | 2.04 |

Rix7 is a type II AAA-ATPase that is required for the assembly of the large ribosomal subunit. Here the authors present the 4.5 Å cryo-EM structure of the Rix7 homohexamer with a polypeptide fragment bound in its central channel and provide insights into the function of Rix7 as a molecular unfoldase

residues are identical. The density for the polypeptide in the channel is ambiguous and was fit was a poly-alanine chain. The presence of nucleotides was determined from a difference map between the nucleotide free hexamer model and the cryo-EM map. The geometry of ATP in the nucleotide binding sites was restrained to the reference structure. Molprobity[68] was used to evaluate the model and the model statistics are listed in Table 1.

## Data availability

All data and constructs used in this study will be made available upon request. Cryo-EM maps and atomic coordinates for the cryo-EM model have been deposited in the EMDB and PDB with accession codes EMD-9063 and PDB 6MAT. The source data underlying Fig. 1b, Fig. 4e, Fig. 6c, and Fig. 6f are provided as a source data file. Other data are available from the corresponding author upon reasonable request.

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

## Acknowledgements

We thank Dr. Scott Williams, Dr. Geoffrey Mueller, and all the members of the Stanley Lab for their critical reading of this manuscript. We would also like to thank all the members of the Molecular Microscopy Consortium at the NIEHS for their help with cryo-EM data collection and processing. We are grateful to Duke University for access to the Krios for data collection. This work was supported by the US National Institutes of Health Intramural Research Program; US National Institute of Environmental Health Sciences (NIEHS) (ZIA ES103247 to R.E.S.).

## Author contributions

Y.H.L., A.L.H., B.L.F., and M.J.B. prepared grids and collected and processed cryo-EM data. J.M.K., Y.H.L., and R.E.S. built and refined the Rix7 model. Y.H.L. and M.S. performed yeast complementation assays, sucrose gradients, and western blots. Y.H.L. and R.E.S. wrote the manuscript and prepared the figures, both of which were approved by all the authors.

## Additional information

**Competing interests:** The authors declare no competing interests.

