## [Peer Review File · Nature Communications]

Reviewers' comments:

Reviewer #1 (Remarks to the Author):

“Unraveling the Mechanism of Substrate Processing by the AAA-ATPase Rix7”

Yu-Hua Lo, Mack Sobhany, Allen L. Hsu, Brittany L. Ford, Juno M. Krahn, Mario J. Borgnia, and Robin E. Stanley

The paper describes the cryo-EM structure of a mutant variant of the AAA-ATPase Rix7 from *Chaetomium thermophilum*.

Rix7 is a Type II AAA-protein that is implicated in the release of Nsa1 and possible other proteins from early nucleolar pre-ribosomal particles. As common for this type of AAA-ATPases, the activity of the protein is strictly dependent on ATP binding to D1 and D2, but ATP hydrolysis in D1 is dispensable for growth. This was shown previously for Rix7 by the Hurt/Kressler group and was confirmed by this manuscript. Additionally, the presented work addresses also the role of other critical residues for the function of AAA-ATPases, like sensor and arginine finger and proved that they are also essential for Rix7.

For structural investigation Lo et al., heterologously expressed the ATPase activity deficient version of the CtRix7 protein (EQ1EQ2 variant) in *Escherichia coli* and purified it using chromatographic methods. Such EQ1EQ2 mutants of AAA-ATPases were shown by many studies to be able to bind the respective substrate proteins but fail to dissociate again due to the inability to hydrolyze nucleotide. It is well established that such mutants act as substrate trap.

The CtRix7 structure presented here is the first structure of this protein and was determined by cryo-EM using a Titan Krios transmission electron microscope equipped with a Falcon III detector to a resolution of about 4.5 Angstroms. It revealed an asymmetric stacked hexameric ring conformation. While the ATPase domains were well defined, the cryo-EM reconstruction unfortunately lacks density for the N-domain. Usually the N-domain of AAA-ATPase represents the regulatory domain of the protein responsible for substrate selection, often in complex with additional cofactors or adaptor proteins. It represents the control gate for the entry into the lumen of the ATPase ring which is ultimately linked with unwinding or degradation of the protein. Although generally more mobile than the rest of the protein and therefore harder to detect in cryo-EM structures, better definition of the N-domain could possibly be obtained by addition of ATPγS as described in Deville et al., 2017. Thus the Authors should also estimate the structure of Rix7 after purification in the presence of this slow hydrolyzing nucleotide. Using state of the art classification software, structure estimation should be possible also for more crude samples of Rix7 after the first purification steps which would limit the required amounts of the expensive nucleotide.

The ATPase domains in the presented structure adopt a staircase like arrangement containing one seam protomer (P6). Surprisingly, the structure contains a peptide string in the central channel of the hexamer. This peptide was suggested by the authors as being part of Rix7 itself or a random peptide co-isolated from *E. coli* during the purification procedure. The substrate is contacted by five of the six protomers and the pore loops grip the substrate in a hand-over-hand mechanism which is reminiscent to processing mechanisms detected previously for other AAA-ATPases like Hsp104 and ClpB in complex with casein as model substrate (Gates et al., 2017; Deville et al., 2017).

AAA-ATPases usually act on very specific substrates and a key factor regulating this specificity is the N-domain which restricts processing of random proteins or peptides. The fact that the heterologously expressed CtRix7 protein encapsulates a peptide in the absence of its native substrate (e.g. Nsa1) implies that the N-domain adopts a non-native conformation.

This represents a clear distinction to the results for HSP104 and ClpB, where a well-established substrate was added *in vitro*. The situation described here for CtRix7 more closely resembles the artificial situation with the archaeal VAT protein, when the N-terminal domain is deleted and the protein unwinds adjacent VATΔN hexamers in some sort of cannibalistic behavior (Ripstein et al., 2017).

A major problem I have with the manuscript of Lo et al., is that the authors try to sell the

structure as true substrate processing intermediate (also implied by the title) while it actually represents an artificial situation. This fact is only mentioned in a short note in the results section (P8 second paragraph, in the middle), which could easily be overlooked and leave the readers with a wrong impression of the substrate processing of Rix7 (e.g. lack of specificity in substrate recognition for Rix7). This is a pity, because without doubt, we could learn a lot about the structure, functioning and substrate processing of Rix7 by the presented work which would be worth publishing in Nature Communications. To prevent misunderstanding, the authors should freely discuss the results in the light that the observed peptide string in the channel is surely not the actual substrate of Rix7.

This fact has to be clearly worked out and discussed in the manuscript and must not be hidden from the readers.

Additional comments: the rationale behind substituting D1 PL-I for D2 PL-I is not clear to me. Of course has the structure (and sequence) of the pore loops evolved depending on the specific environment and geometry within the D1 and D2 domain. The observed growth defect is therefore not really surprising. Alanine substitution as performed for D2 would have been at least equally informative.

Moreover, it is clear that the exchange of (the small) serine 280 for the bulky tyrosine residue in the narrow channel between the protomers will have a pronounced effect on translocation of any peptide. To infer that this residue might have a special role in substrate selection in vivo from this experiment is not backed up by the presented data at all and pure speculation. All available data from AAA-ATPases I am familiar with suggest that the substrate selection is performed by the N-domain, possibly in conjunction with additional factors, but surely not by the pore loop residues. These loop residues have to accommodate a number of different amino acid residues during translocation of a single polypeptide chain and therefore cannot be selective or involved in substrate discrimination as stated by the authors.

Minor issues: The last two sentences of the abstract do not reflect what was shown here and have to be changed

The term Introduction is missing.

P4, 1. Paragraph last sentence: Nsa1 stays on the particle under rix7 mutant conditions, this strongly suggests that it is indeed the release substrate and not a cofactor.

P4, 2. Paragraph last sentence: The cited work from the Rapaport group refers to Cdc48 and not to p97.

P4 last sentence: avoid the term substrate here, because substrate would be a small, clearly defined set of proteins for Rix7 and not an E. coli protein.

P6: second paragraph: The effects on the polysome profiles of the rix7 mutants are not very pronounced, the authors should state somewhere (material and methods section) how long the cells were incubated in the presence of DOX and to which density the cells were initially inoculated for the individual mutants.

P9 heading: cerevisiae

P13 (Discussion), first paragraph, last sentence: I do not see how the data presented here distinguish Rix7 from other substrate threading mechanisms. Definitely it is not ATP hydrolysis (which is required for all AAA-ATPases).

P13: "Our data revealed the importance of a well conserved serine residue from the D1 PL-1 signature motif that likely only allows access of specific substrate(s) the central channel" (see my comments above).

P14 1.sentence: Cdc48 not p97

P14, second paragraph: Although alpha 7 could indeed increase the stability of the D1 hexameric ring as the authors speculate, this was not shown experimentally and therefore cannot be stated as a fact.

In the same paragraph: The issue that the authors could not detect the N-domain in their cryo-EM reconstruction does not mean that the N-domain does not exist or does not have any function and of course does not mean that the D1-domain overtakes its function in substrate recruitment/selection. This has to be changed also in P22 (legend to Figure 6).

Reviewer #2 (Remarks to the Author):

This is a clear and well written manuscript about the molecular mechanisms of AAA-ATPase Rix7 as an unfoldase. This protein has been demonstrated to have a role in the assembly of the large ribosomal subunit in yeast. Its role is essential. The proposed role for this enzyme is to use the energy from ATP hydrolysis to drive the release of assembly factors from the immature 60S subunits. In particular, it has been proposed that Rix7 drive the release of assembly factor Nsa1. This manuscript describes the structure of Rix7 by cryo-EM and addresses the question on how Rix7 pull on substrates to drive their removal from the immature 60S particles. The study also identifies key residues that are required within D1 and D2 AAA domains for ribosome assembly and cell viability.

I think the manuscript is acceptable for publication but before authors should address these concerns below:

1. I gather from the images and numbers shown in Suppl. Fig. 3 that an important problem in the behaviour of Rix7 in the cryo-EM grids was preferential orientation. It seems that top and slightly oblique views were much more abundant than side views in the dataset. Based on the numbers they seem to be at a ratio of ~52:1 (top:side views). This ratio of orientation bias should produce a distorted 3D model, anisotropic resolution, stretching in one direction and appearance of flat densities for alpha helices in the protein. These distortions compounded with the obtained average resolution of the cryo-EM map above 4Å should make the production of the atomic model through modeling procedures highly unreliable if not impossible. It is my understanding that the 3D reconstruction algorithms used in Relion are not able to take care of such an uneven orientation distribution without modifications. Have the authors introduce intermediate processing steps not described in the methods to compensate for the orientation bias. Perhaps authors should consider using the scripts from Michael Cianfrocco (<https://github.com/leschzinerlab/Relion>) to randomly remove particles from the most populated orientations. There has been also many methods described to change this non-optimal behaviour of the proteins in the grid, including addition of detergents, chemical treatments of the grids, graphene grids, etc. I wonder if the authors attempted any of these methods at the sample prep optimization stages of the project to overcome this issue.

2. The presented cryo-EM map has not been sufficiently validated. That would not be an issue if the resolution obtained was more in the 3Å range. At that resolution level visualization of side chains provide sufficient confidence that the atomic model is correct. In this case, because the resolution achieved (~4.5Å) does not allow to resolve side chains well, and the problem is aggravated by a preferential orientation issue some validation experiments are recommended. First, the authors use the routines in EMAN2 to produce the initial model to prime 3D classifications and refinements. This is highly risky for this particular sample. EMAN2 tutorial described how "strongly preferred orientation, especially if this is combined with a low symmetry particle, there may not be enough information to produce an unambiguous starting model. For most structures, there are a number of 'local minima' in the energy space. What that means is, there are a number of incorrect structures which can still appear to agree fairly well with the input data. So, some fraction of the answers you get out are likely to be bad starting models. The severity of this problem varies considerably with the shape of the molecule and the amount of orientation coverage you have. GroEL, with its strongly preferred orientation and nearly square shape in the side view, is actually among the most difficult structures to produce a good starting model for." This is very much the case here for Rix7. The authors should consider perhaps re-run the data processing using p97 as an initial model. Perhaps also use some of the newer ab initio 3D structure determination algorithms that have been recently implemented in Relion, CryoSPARC or RAMSAC in Scipion. How do the ab initio maps obtained with these algorithms compare with the one obtained with EMAN2 (not ideal for the sample at hand). Optimally, in a high-risk sample such as that one in this study a tilt validation assay should be performed for structure validation. As I

am certain the authors know, in this assay images are collected at 0 degrees and typically 10-20 degrees tilt. Box out tilt pairs of particles are then run through a tilt validation procedure against the final cryo-EM map.

3. It is unclear to this reviewer why the authors decided to run their autopicking routines with a gaussian blob? Relion offers the possibility to generate templates by manually picking a few thousand of images. Based on Suppl. Fig 3, the 2D classification step throw away ~70% of the picked particles. I think that result is a clear sign that the autokicking step was not well optimized. Also, a gaussian blob certainly will not favor picking side-view particles. They are very different from a gaussian blob. Consequently, it is likely that the possible orientation bias existing with Rix7 was aggravated with the autokicking approach undertaken by these authors. Authors should at least explain the rationale for their autopicking approach.

4. In the class2D steps: How many classes were requested for the first step of classification before the dataset was split into top and side views? Did they only request 10-15 classes as in the second step? Why so few? For a dataset of this size with over half a million particles a much larger number of classes is typically used (~100-150).

5. A limitation of the obtained structure was that the NTD of Rix7 protomer was not visible, presumably due to flexibility. Did the authors attempted focus classification with signal background subtraction in these regions to attempt describe these structural motifs?

6. Regarding the density in the axial channel presumably representing a client protein being pulled out and threaded through the pore is an interesting finding. However, if I understood correctly the description from the authors, a substrate would be required to always be threaded in the same way through the pore in order to produce a defined density. Are the authors proposing that as the substrate is threaded is handed over from the pore loop in one protomer to the next in a specific sequential manner? Are they suggesting that because of the asymmetric nature of the hexamers threaded substrate always enter and progress through the axial pore in a defined sequence through specific protomers? This would be as opposed to what is proposed in the manuscript that there are 6 equivalent positions in the ring that the substrate could use as it is translocated through. In this last case, cryo-EM would not provide a defined density but rather a fragmented density resulting from the averaging of particles with substrates bound on all six positions. The authors should elaborate about this point in their discussion.

Reviewer #3 (Remarks to the Author):

What are the major claims of the paper?

The manuscript by Lo et al., "Unraveling the Mechanism of Substrate Processing by the AAA-ATPase Rix7" describes in details the structure of the AAA+ C.thermophilum Rix7, an ATPase involved in the production/maturation of the large 60S ribosomal subunit by removal of the assembly factor Nsa1. The authors present the structure of the hexameric Rix7 assembly in complex with a threaded substrate. They therefore determine that the protein is definitively an unfoldase. They deduce the unfolding mechanism from the structure and describe the specific and different roles of the two AAA domains.

Are the claims novel? If not, please identify the major papers that compromise novelty

The manuscript contains novel results but the major claims are actually not so novel. Here below I explain more in details my point of view hoping to help the authors to bring forward the more novel findings contained in the manuscript.

The structure of Rix7 is new as, apart from the N-terminal NMR structure of NVL2 (Fujiwara et al, JBC 2011) and the AAA-2 of NVL2 (pdb code 2x8a (3 helices Nter) no other structural information about Rix7/NVL2 was known. However, the oligomeric arrangement of Rix7 is almost identical to

that of other AAA+ unfoldases recently studied by EM. A number of two-cassette AAA+ unfoldase structures have been solved in the last years and they all show the same unfolding mechanism with one "seam" subunit of the AAA+ rings that perform the pulling force. All these structures are also all analysed in a similar way with particular emphasis to the pore substrate-binding loops organisation (Fig 4a-c in this manuscript) and the relative position of the 6 subunits within the hexameric ring (Fig 3 in this manuscript). Even though it is correct and thorough to perform all the analysis of the canonical AAA+ features to find whether or not the Rix7 looks similar to the already known AAA+ unfoldases, once found that the structural features of subunits organisation, pore loops and nucleotides pockets are actually very similar to all other AAA+ unfoldase structure solved so far, the authors should have acknowledge it more and stress instead more the discovery of structural features specific of Rix7 and indeed novel and interesting.

The authors analyse two specific structural features of Rix7, the well-conserved serine in the PLI loop of AAA1 and the post alpha7 insertion, that should be given more emphasis in the text and figures as they would strengthen the manuscript and make it more interesting for researcher interested in the specific biological function of Rix7, ribosome large subunit maturation. AAA proteins have very well conserved motors, but perform extremely different biological activities, so a lot of novelty is in the details that differentiate one motor from the other.

Will the paper be of interest to others in the field?

The paper will be of interest to others in the field of structural biology specifically in the field of ribosome maturation, chaperones in general and of course AAA proteins. The paper could also be of interest from an evolution point of view as it adds information about a specific class within the large AAA+ superfamily.

Will the paper influence thinking in the field?

This is hard to tell. Might affect the thinking in terms of understanding the steps of ribosome formation.

Are the claims convincing? If not, what further evidence is needed?

The claims are convincing, they are strongly based on previous literature and find nice corroboration in the structural visualisation. The combination of functional in vivo experiments with the structural analysis is well performed.

Are there other experiments that would strengthen the paper further? How much would they improve it, and how difficult are they likely to be?

Some analysis of the N-terminus of Rix7 would be appreciated. More specific comments are found below.

Are the claims appropriately discussed in the context of previous literature?

The authors cite the existing literature and analyse the structure of hexameric Rix7 in a fairly canonical way complying with all the requirements so far established in the field (analysis of AAA defining motifs such as walker A, B, R-finger, nucleotide binding, substrate binding). They refer to the literature correctly, but they tend to tune down the similarities of their structure to all the deposited and published maps of similar unfoldases. The authors should not worry about being completely honest about the extreme similarity of Rix7 with other AAA+ structure as Rix7 is a different protein, performs a different activity and in fact the paper also contains interesting and novel information about the specific structural features of Rix7.

If the manuscript is unacceptable in its present form, does the study seem sufficiently promising that the authors should be encouraged to consider a resubmission in the future?

The manuscript is already in a form acceptable for publication, but changes (details below) might improve it.

Is the manuscript clearly written? If not, how could it be made more accessible?

The manuscript is clearly written.

Could the manuscript be shortened to aid communication of the most important findings?
The manuscript is already in a fairly compact form.

Have the authors done themselves justice without overselling their claims?
The authors have done themselves justice, sometimes up-tuning and sometimes down-tuning the novelty of some results.

Have they been fair in their treatment of previous literature?
Yes, they have.

Have they provided sufficient methodological detail that the experiments could be reproduced?
For the cryo-EM parts, I think they did. Some choices should be justified (see details below).

Should the authors be asked to provide further data or methodological information to help others replicate their work? (Such data might include source code for modelling studies, detailed protocols or mathematical derivations).
The authors should have already deposited the map and the pdb models and the EMDB and PDB codes should be stated in the manuscript.

Comments along the text and figures :

In the introduction the authors limit their structural comparison to NSF, p97 and Pex1/Pex6. This, even though formally correct as these proteins are the closest relative to Rix7 within the AAA+ superfamily, it is a bit misleading as there are nowadays various cryo-EM maps of two-cassette AAA+ proteins. The article refers to these works later on, but in the introduction only NSF, p97 and Pex1/Pex6 are mentioned. It appears as a way a bit forced of adding novelty to the findings. The structure of VAT, which belongs to the same class and has a substrate-bound must also be mentioned in the introduction for completion. Likewise the introduction states "The structure reveals an unexpected asymmetric configuration ...". Asymmetric configurations of AAA+ rings are not unexpected anymore. Also the fact that 5 out of 6 subunits grab the substrate is not unexpected. The authors should tune down the novelty of the structural arrangement of Rix7 in the last paragraph of the introduction. Tuning it down should not undermine the novelty and beauty of the Rix7 structure anyway because it is true that it is for the first time Rix7 is visualised and clearly shown to be a unfoldase. The authors should include at the end of the introduction the results about the specific features of Rix7, they are in the discussion. I think that these results are novel and add knowledge to the process of understanding how the same AAA molecular motor evolved to perform such different activities. This is one of the main open questions in the AAA field and the author could stress their contribution to it in the manuscript.

Paragraph "ATPase activity of Rix7 modulates 60S subunit formation". Elegant experimental setup with the regulable Rix7 endogenous expression for the analysis of the various mutants in vivo. It is not clear how many times were the sucrose gradient experiments repeated, should be said.

Paragraph "Asymmetric hexameric architecture of Rix7". For the model building, did the authors check the structure of the AAA2 domain of NVL2 (pdb code 2x8a)? It is not published but the coordinates are deposited. How similar is it to the manuscript model? The N-terminus of Rix7 is not visible at all. Did the authors try to analyse via focused classification the terminal region of the molecule? The sentence "The individual AAA domains all superimpose well with one another but there are differences in the relative orientations between D1 and D2 AAA domains, which gives rise to the spiral arrangement of the protomers" is unclear. From this sentence it seems as if the small and large subdomains of each individual AAA superimpose well which is not the case as later explained in Figure 5 and SuppFig 5b. Moreover, difference in the relative orientation of D1 and D2 give rise to two different types of spiral arrangements in the D1 and D2 ring, not to the spiral as such. One could have difference in the relative orientation of D1 and D2 and have a planar ring.

Paragraph "Rix7 threads substrates through its central pore". The fact that Rix7 contains a substrate of unknown origin indicates that it is fairly non-specific as unfoldase. Can the author comments on it? The paragraph that describes the particularity and substrate contacts of the pore loops of Rix7 even when the aromatic residue in the loop triad for substrate binding is lacking, is interesting and novel and should be put more in light. The authors could make a dedicated panel showing the densities and the contact in between h-S-G and the substrate. This novel result shock be also spelt out in the introduction and abstract in my opinion.

Paragraph "Rix7 pore loops are essential in *S.cerevisiae*. The hypothesis that the pore loops of Rix7 play a role in substrate selection is interesting. It is a novel point of the paper. Is it possible to run in vitro substrate affinity tests using putative substrates? Rix7 clearly binds to some substrate here during purification. The density is not good enough to determine the aa sequence of the substrate bound, but did the authors try to run mass spectrometry experiments to attempt determining the identity of the bound substrate? Do the author think that an adaptor protein is also playing a role in vivo?

Paragraph "The seam promoter and the nucleotide-bound state". Is there enough resolution to claim that the walker A P loop adopts different orientations? This is difficult to judge without having the map. The authors should show a figure with the density map for this area. E.g. Figure 5 c should show the actual densities.

Paragraph "Insertion following alpha7 in the D1 and D2 domains". The analysis of the insertion alter alpha 7 is interesting and novel. The authors should in my opinion stress it in the introduction/abstract and make SuppFig 6 (or a variant of it) a main figure. Are there some mutation that the authors could make on the post alpha 7 insertion to somehow "block" the Nterminus (for cryo-EM studies)? Of all the 3D classes obtained none, not even at low resolution and threshold included the N-terminus?

Discussion. The claim that Rix7 shows a unique combination of substrate threading by D1 and processing translocation by D2 is not sufficiently funded by the structure nor explained with words. Should be clarified.

-Methods

Did the authors try to perform 3D classification as first step, even before 2D? They state they refined the model in reciprocal space. Can they confirm they did not touch the map? The map and pdb must be deposited.

More general questions. In a previous work the authors describe interaction of WDR74 and NVL2 via the AAA1 ring. Looking at the Rix7 structure can they make any comment/rational about the nature of this interaction?

From the structure and the biochemical experiments here performed any idea on whether Nsa1 is a cofactors rather than a substrate?

Figures:

Figure1 and 2 ok

Figure 3 ok , but it would be nice to show an insight with the density for the nucleotide (if/when visible)

Figure 4 a) this panel does not add more information than the panel b as it is generated anyway from the model. Might be nicer to show the actual density from the EM map.

Figure 5 c should be changed and real density of the walker A areas should be shown rather than the model.

Supp Fig1 and 2 ok

Supp Fig3 Why was C2 symmetry applied in the first place?

Supp Fig4 the FSC of masked, unmasked, phase randomised maps should be shown. The angular distribution still shows some preferential orientation Did the authors do anything to balance the views during image processing? If so, they should describe.

Supp Fig5 ok

Supp Fig6 with changes should become a main figure.

Reviewers' comments (*italics*) and our reply (**red**).

Reviewer #1 (Remarks to the Author):

The paper describes the cryo-EM structure of a mutant variant of the AAA-ATPase Rix7 from Chaetomium thermophilum. Rix7 is a Type II AAA-protein that is implicated in the release of Nsa1 and possible other proteins from early nucleolar pre-ribosomal particles. As common for this type of AAA-ATPases, the activity of the protein is strictly dependent on ATP binding to D1 and D2, but ATP hydrolysis in D1 is dispensable for growth. This was shown previously for Rix7 by the Hurt/Kressler group and was confirmed by this manuscript. Additionally, the presented work addresses also the role of other critical residues for the function of AAA-ATPases, like sensor and arginine finger and proved that they are also essential for Rix7. For structural investigation Lo et al., heterologously expressed the ATPase activity deficient version of the CtRix7 protein (EQ1EQ2 variant) in Escherichia coli and purified it using chromatographic methods. Such EQ1EQ2 mutants of AAA-ATPases were shown by many studies to be able to bind the respective substrate proteins but fail to dissociate again due to the inability to hydrolyze nucleotide. It is well established that such mutants act as substrate trap. The CtRix7 structure presented here is the first structure of this protein and was determined by cryo-EM using a Titan Krios transmission electron microscope equipped with a Falcon III detector to a resolution of about 4.5 Angstroms. It revealed an asymmetric stacked hexameric ring conformation.

We thank the reviewer for their very thorough and careful review of our manuscript. We responded to each point raised by the reviewer below.

Point 1.1) *While the ATPase domains were well defined, the cryo-EM reconstruction unfortunately lacks density for the N-domain. Usually the N-domain of AAA-ATPase represents the regulatory domain of the protein responsible for substrate selection, often in complex with additional cofactors or adaptor proteins. It represents the control gate for the entry into the lumen of the ATPase ring which is ultimately linked with unwinding or degradation of the protein. Although generally more mobile than the rest of the protein and therefore harder to detect in cryo-EM structures, better definition of the N-domain could possibly be obtained by addition of ATP γ S as described in Deville et al., 2017. Thus the authors should also estimate the structure of Rix7 after purification in the presence of this slow hydrolyzing nucleotide. Using state of the art classification software, structure estimation should be possible also for more crude samples of Rix7 after the first purification steps which would limit the required amounts of the expensive nucleotide.*

We thank the reviewer for the suggestion to collect a dataset of the Rix7 EQ1EQ2 mutant in the presence of ATP γ S. We think that this is a wonderful suggestion and we actually collected this dataset prior to our submission of the original version of the manuscript. Data was collected using a Titan Krios transmission electron microscope operated at 300 keV and equipped with a Falcon 3EC detector. A total of 915 micrographs were collected, from which a subset of 71,465 particles were used to obtain a 5.9 Å reconstruction of Rix7. Unfortunately, we could not observe any additional density for the NTD within this map. We also attempted to collect data using wild-type (WT) Rix7 in the presence of different nucleotides (ATP, ADP, and ATP γ S), however WT Rix7 is very unstable compared to the Walker B mutant and we have yet to establish a protocol that results in grids that are suitable for data collection. This work is still ongoing in the lab and while we think additional nucleotide bound states are beyond the scope

of this manuscript they will be extremely important for understanding ATP hydrolysis dependent structural rearrangements within Rix7.

The reviewer makes an excellent point about the importance of the regulatory NTD (N-terminal domain) of type II ATPases. One thing that we failed to highlight in our original submission was how different the NTD of Rix7 is from other ATPases. In contrast to type II ATPases such as p97/Cdc48/VAT and the ClpB/Hsp100 family, the NTD of Rix7 contains a large region of predicted disorder. Shown below is the disorder prediction by IUPred (Dosztányi et al. 2005. *Bioinformatics* 21: 3433-3434) and GlobPlot2.3 (Linding et al 2003. *NAR* 31(13): 3701-3708) for Rix7, NVL2 (mammalian homologue of Rix7), p97, and ClpB indicating the disordered nature of the second half (residues 100-200) of the Rix7/NVL2 NTD (indicated by black arrow in below figures).

Results from disorder prediction server by IUPred:

5FTK_Human_p97

5OFO_Ec_ClpB

6MAT_Ct_Rix7

Human_NVL2

Results from disorder prediction server by GlobPlot2.3:

5FTK_Human_p97

5OFO_Ec_ClpB

6MAT_Ct_Rix7

Human_NVL2

Based upon these disorder predictions it is not unexpected that we cannot observe density for the disordered NTD of Rix7. Recently, Bodnar et al reported a complex structure of Cdc48 with its cofactor Ufd1–Npl4 which suggested that ATP and cofactor binding cooperate to move the NTDs into the up conformation (Bodnar et. al., Nature Structural & Molecular Biology, 2018, 25, 616–622). We therefore assume that specific cofactors or adaptor proteins of Rix7 will likely be necessary for regulating the intrinsically unstable region of the NTD. We further elaborate on the disordered Rix7 NTD in both the results (page 7, second paragraph) and discussion (page 13, last paragraph).

Point 1.2) The ATPase domains in the presented structure adopt a staircase like arrangement containing one seam protomer (P6). Surprisingly, the structure contains a peptide string in the central channel of the hexamer. This peptide was suggested by the authors as being part of Rix7 itself or a random peptide co-isolated from *E. coli* during the purification procedure. The substrate is contacted by five of the six protomers and the pore loops grip the substrate in a hand-over-hand mechanism which is reminiscent to processing mechanisms detected previously for other AAA-ATPases like Hsp104 and ClpB in complex with casein as model substrate (Gates et al., 2017; Deville et al., 2017). AAA-ATPases usually act on very specific

substrates and a key factor regulating this specificity is the N-domain which restricts processing of random proteins or peptides. The fact that the heterologously expressed CtRix7 protein encapsulates a peptide in the absence of its native substrate (e.g. Nsa1) implies that the N-domain adopts a non-native conformation. This represents a clear distinction to the results for HSP104 and ClpB, where a well-established substrate was added *in vitro*. The situation described here for CtRix7 more closely resembles the artificial situation with the archaeal VAT protein, when the N-terminal domain is deleted and the protein unwinds adjacent VATDN hexamers in some sort of cannibalistic behavior (Ripstein et al., 2017). A major problem I have with the manuscript of Lo et al., is that the authors try to sell the structure as true substrate processing intermediate (also implied by the title) while it actually represents an artificial situation. This fact is only mentioned in a short note in the results section (P8 second paragraph, in the middle), which could easily be overlooked and leave the readers with a wrong impression of the substrate processing of Rix7 (e.g. lack of specificity in substrate recognition for Rix7). This is a pity, because without doubt, we could learn a lot about the structure, functioning and substrate processing of Rix7 by the presented work which would be worth publishing in Nature Communications. To prevent misunderstanding, the authors should freely discuss the results in the light that the observed peptide string in the channel is surely not the actual substrate of Rix7. This fact has to be clearly worked out and discussed in the manuscript and must not be hidden from the readers.

The reviewer raises a very important issue about the polypeptide that we observe in the central channel and the reviewer is absolutely correct that this peptide likely does not represent a true substrate of Rix7. We were very surprised when we solved the structure to observe density within the central channel as we did not add a substrate as was done in the recent HSP104 and ClpB structures referenced above. We apologize if our initial manuscript was misleading. To avoid misleading reviewers we have changed the text in several places, including the abstract to make it clear that the polypeptide observed within the central channel is not an actual substrate. We have changed the title of the manuscript to "Cryo-EM Structure of the Essential Ribosome Assembly AAA-ATPase Rix7". We also elaborate in the discussion about the implications of the polypeptide as a substrate mimic. Despite trapping Rix7 with an unknown polypeptide our Rix7 structure reveals important details about the function of Rix7 and how it processes substrates. This is the first structure of the Rix7 homohexamer and the first indication that Rix7 functions as a molecular unfoldase. Future studies will be needed to establish substrate selectivity of Rix7.

Point 1.3) Additional comments: the rationale behind substituting D1 PL-I for D2 PL-I is not clear to me. Of course has the structure (and sequence) of the pore loops evolved depending on the specific environment and geometry within the D1 and D2 domain. The observed growth defect is therefore not really surprising. Alanine substitution as performed for D2 would have been at least equally informative. Moreover, it is clear that the exchange of (the small) serine 280 for the bulky tyrosine residue in the narrow channel between the protomers will have a pronounced effect on translocation of any peptide. To infer that this residue might have a special role in substrate selection *in vivo* from this experiment is not backed up by the presented data at all and pure speculation. All available data from AAA-ATPases I am familiar with suggest that the substrate selection is performed by the N-domain, possibly in conjunction with additional factors, but surely not by the pore loop residues. These loop residues have to accommodate a number of different amino acid residues during translocation of a single polypeptide chain and therefore cannot be selective or involved in substrate discrimination as stated by the authors.

We absolutely agree with the reviewers comments that pore loop residues have to accommodate a number of different amino acid residues during translocation of a single

polypeptide chain and therefore might not be selective or involved in substrate discrimination. We have modified the text accordingly and do not refer to the D1 PL-I as being directly involved in substrate selection. We do however think it's quite plausible that other regions of the D1 domain are important for substrate selection. We previously showed that WDR74, the mammalian homologue of Nsa1 binds to the D1-AAA domain of Rix7/NVL2 and not the NTD (Lo et al Structure, 2017). This is in contrast to other type II ATPases which utilize their NTDs for substrate selection and co-factor binding. While it is yet to be fully established if Nsa1/WDR74 is a substrate of Rix7/NVL2, the NTD of Rix7 is dispensable for the association of the mammalian homologues of these two proteins.

A unique and novel feature of Rix7 is the absence of the aromatic-hydrophobic pore loop motif within the D1 domain. Our reasoning for substituting the D2 PL-I residues with the D1 PL-I residues was based on similar studies done in Cdc48 and p97. Rothballer *et al* (FEBS Letters 2007) demonstrated that both removal of the NTD and introduction of a YY motif (D2 PL-I motif in p97) in the D1 PL-I turned p97 into a non-specific molecular unfoldase *in vitro*. Recent studies by Esaki *et al* (Scientific Reports, 2017) revealed that introduction of aromatic residues to the Cdc48 PL-I is lethal in yeast. We agree that the S280Y mutation was quite severe but an important mutation to make, since Trp is a conserved amino acid found in pore-loops of the unfoldase family. We have now included a less severe mutation (S280A) and show that this mutation causes a moderate growth defect in yeast, further emphasizing the importance of the S280 residue in Rix7. Please see Figure 4e in our updated manuscript.

Minor issues:

Point 1.4) *The last two sentences of the abstract do not reflect what was shown here and have to be changed*

We have modified the abstract as requested. The last three sentences of the abstract have been replaced with the following:

Here we report the cryo-EM reconstruction of the tandem AAA domains of Rix7 which form an asymmetric stacked homohexameric ring. We trapped Rix7 with a polypeptide in the central channel, revealing Rix7's role as a molecular unfoldase. The structure establishes that type II AAA-ATPases lacking the canonical unfoldase residues within the first AAA domain can engage a substrate throughout the entire central channel. The structure also revealed that Rix7 contains unique post- α 7 insertions within both AAA domains important for its function.

Point 1.5) *The term Introduction is missing.*

According to the *Nature Communications* "Guide to Authors" the term Introduction should not be included.

Point 1.6) *P4, 1. Paragraph last sentence: Nsa1 stays on the particle under rix7 mutant conditions, this strongly suggest that it is indeed the release substrate and not a cofactor.*

While we absolutely agree with the reviewer that there is compelling evidence that Nsa1 is a substrate of Rix7, this has not been conclusively shown. Reconstitution studies will be needed to fully address this question.

Point 1.7) P4, 2. Paragraph last sentence: *The cited work from the Rapaport group refers to Cdc48 and not to p97.*

We thank the reviewer for pointing out our mistake and we have corrected it in the text.

Point 1.8) P4 last sentence: *avoid the term substrate here, because substrate would be a small, clearly defined set of proteins for Rix7 and not an E. coli protein.*

As mentioned above under point 1.2 we now refer to this as a polypeptide.

Point 1.9) P6: second paragraph: *The effects on the polysome profiles of the rix7 mutants are not very pronounced, the authors should state somewhere (material and methods section) how long the cells were incubated in the presence of DOX and to which density the cells were initially inoculated for the individual mutants.*

We have included the following additional information in the materials and methods section (see Page 19, first paragraph):

Starter cultures of transformed *tetO₇-Rix7* (Supplementary Table 2) were grown at 30°C in the presence of DOX (120 µg/ml) for 16 hours. 1L cultures of YPD and DOX (120 µg/ml) were inoculated to an OD of 0.05 and then grown at 30°C to an of OD ~0.6. Cycloheximide (0.1 mg/ml) was added to the cultures and the cultures were incubated for 5 minutes on ice before harvesting cells by centrifugation. Sucrose gradients and polysome profiling were performed as described in Pillon et al.

Point 1.10) P9 heading: *cerevisiae*

We thank the reviewer for pointing out this typo; it has been corrected (Page 10, heading).

Point 1.11) P13 (Discussion), first paragraph, last sentence: *I do not see how the data presented here distinguish Rix7 from other substrate threading mechanisms. Definitely it is not ATP hydrolysis (which is required for all AAA-ATPases).*

We agree that this sentence was confusing and we have removed this sentence from the manuscript (Page 14, first paragraph).

Point 1.12) P13: *"Our data revealed the importance of a well conserved serine residue from the D1 PL-1 signature motif that likely only allows access of specific substrate(s) the central channel" (see my comments above).*

As discussed above under Point 1.3 we have modified the text as requested.

Point 1.13) P14 1.sentence: *Cdc48 not p97*

We thank the reviewer for pointing out our mistake and we have corrected it in the text (Page 15 second paragraph).

Point 1.14) P14, second paragraph: Although alpha 7 could indeed increase the stability of the D1 hexameric ring as the authors speculate, this was not shown experimentally and therefore cannot be stated a fact.

The reviewer makes an important point that we did not show data to support this claim and it is purely a hypothesis. Therefore we have removed this claim from the manuscript. We made deletions to the post alpha 7 insertion from both the D1 and D2 domains for yeast complementation assays and now include this data in the updated manuscript (Figure 6c and 6f). This data does not indicate if alpha 7 enhances stability but it highlights the importance of the post alpha 7 insertions *in vivo* (please see page 12, last paragraph).

Point 1.15) In the same paragraph: The issue that the authors could not detect the N-domain in their cryo-EM reconstruction does not mean that the N-domain does not exist or does not have any function and of course does not mean that the D1-domain overtakes its function in substrate recruitment/selection. This has to be changed also in P22 (legend to Figure 6).

We agree with the reviewer, just because we could not observe the NTD does not mean it is not important. Others have shown that the NTD of Rix7 is essential and important for nuclear/nucleolar localization, however its molecular function remains unclear. As mentioned above under Point 1.3 the NTD of NVL2 (mammalian homologue of Rix7) is not required for association with WDR74 (mammalian homologue of Nsa1), suggesting that the D1 domain of Rix7 may play the role in substrate recruitment. We have changed the figure legend (this is now Figure 7) to state “a target substrate is engaged by the D1 domain.”

Reviewer #2 (Remarks to the Author):

This is a clear and well written manuscript about the molecular mechanisms of AAA-ATPase Rix7 as an unfoldase. This protein has been demonstrated to have a role in the assembly of the large ribosomal subunit in yeast. Its role is essential. The proposed role for this enzyme is to use the energy from ATP hydrolysis to drive the release of assembly factors from the immature 60S subunits. In particular, it has been proposed that Rix7 drive the release of assembly factor Nsa1. This manuscript describes the structure of Rix7 by cryo-EM and addresses the question on how Rix7 pull on substrates to drive their removal from the immature 60S particles. The study also identifies key residues that are required within D1 and D2 AAA domains for ribosome assembly and cell viability.

I think the manuscript is acceptable for publication but before authors should address these concerns below:

We thank the reviewer for their support of our manuscript for publication in *Nature Communications*. Following is our response to their specific concerns.

Point 2.1) *I gather from the images and numbers shown in Suppl. Fig. 3 that an important problem in the behaviour of Rix7 in the cryo-EM grids was preferential orientation. It seems that top and slightly oblique views were much more abundant than side views in the dataset. Based on the numbers they seem to be at a ratio of ~52:1 (top:side views). This ratio of orientation bias should produce a distorted 3D model, anisotropic resolution, stretching in one direction and appearance of flat densities for alpha helices in the protein. These distortions compounded with the obtained average resolution of the cryo-EM map above 4Å should make the production of the atomic model through modeling procedures highly unreliable if not impossible. It is my understanding that the 3D reconstruction algorithms used in Relion are not able to take care of*

such an uneven orientation distribution without modifications. Have the authors introduce intermediate processing steps not described in the methods to compensate for the orientation bias. Perhaps authors should consider using the scripts from Michael Cianfrocco (<https://github.com/leschzinerlab/Relion>) to randomly remove particles from the most populated orientations. There has been also many methods described to change this non-optimal behaviour of the proteins in the grid, including addition of detergents, chemical treatments of the grids, graphene grids, etc. I wonder if the authors attempted any of these methods at the sample prep optimization stages of the project to overcome this issue.

We thank the reviewer for their suggestions on how to overcome orientation bias. We tried different type of grids (QUANTIFOIL R2/2 Au grid 200, QUANTIFOIL R1.2/1.3 Cu grid 300 +2nmC, C-flat 1.2/1.3-4Cu-50 Protochips), several detergents with various concentration (tween 20, NP40, Glucopyranoside) and chemical treatments of the grids but none were helpful in overcoming orientation bias for Rix7. We therefore took a brute force approach to collect enough images to gain a sufficient number of side-view particles for 3D reconstruction. The reviewer is correct that not having enough side views will produce a distorted and anisotropic 3D model. Despite the preferred orientation bias we were still able to get enough coverage of different views of Rix7 to generate a map with isotropic resolution. Shown below are additional views of the map illustrating that it contains isotropic features. We also added a panel to Supplementary Figure 4b to illustrate the map of one protomer from two different directions to demonstrate that the map is isotropic in both directions. The map has already been deposited in the EMBD (Entry ID EMD-9063) but we would be more than happy to provide the map to the reviewer so that they can directly inspect the quality of the map.

Point 2.2) The presented cryo-EM map has not been sufficiently validated. That would not be an issue if the resolution obtained was more in the 3Å range. At that resolution level visualization of side chains provide sufficient confidence that the atomic model is correct. In this case, because the resolution achieved (~4.5Å) does not allow to resolve side chains well, and the problem is aggravated by a preferential orientation issue some validation experiments are recommended. First, the authors use the routines in EMAN2 to produce the initial model to prime 3d classifications and refinements. This is highly risky for this particular sample. EMAN2 tutorial described how “strongly preferred orientation, especially if this is combined with a low symmetry particle, there may not be enough information to produce an unambiguous starting model. For most structures, there are a number of ‘local minima’ in the energy space. What that means is, there are a number of incorrect structures which can still appear to agree fairly well with the input data. So, some fraction of the answers you get out are likely to be bad starting models. The severity of this problem varies considerably with the shape of the molecule and the amount of orientation coverage you have. GroEL, with its strongly preferred orientation and

nearly square shape in the side view, is actually among the most difficult structures to produce a good starting model for.” This is very much the case here for Rix7. The authors should consider perhaps re-run the data processing using p97 as an initial model. Perhaps also use some of the newer ab initio 3D structure determination algorithms that have been recently implemented in Relion, CryoSPARC or RAMSAC in Scipion. How do the ab initio maps obtained with these algorithms compare with the one obtained with EMAN2 (not ideal for the sample at hand). Optimally, in a high-risk sample such as that one in this study a tilt validation assay should be performed for structure validation. As I am certain the authors know, in this assay images are collected at 0 degrees and typically 10-20 degrees tilt. Box out tilt pairs of particles are then run through a tilt validation procedure against the final cryo-EM map.

We understand the concerns regarding this point and thank the reviewer for the opportunity of providing clarification. The reviewer is correct in pointing out that molecular images at low signal to noise ratio can be back projected to produce incorrect solutions to the projection problem (i.e. incorrect maps). It is important to clarify that there are two sources of confusion that add up to produce incorrect maps in intermediate resolution. The first one is related to the method used to produce an initial map and is particularly insidious in methods that make use of back projection. The back-projection problem is ill-posed as it may produce a number of alternative solutions due to the presence of “phantom volumes” (for a detailed explanation on this see Penczek, Methods Enzymol. 2010; 482: 1–33.). The second problem, referred in the EMAN tutorial as the “a number of ‘local minima’ in the energy space”. A fundamental problem in image processing for cryo-EM is the comparison of each experimental molecular image with the reference during refinement. The search for the correct answer involves the minimization of a function with at least five degrees of freedom: the image needs to be shifted in X and Y to center it relative to the reference, and the reference needs to be rotated about three axes in order to find the orientation that gave origin to the projection. This precludes the possibility of performing an exhaustive search of the “energy space” if the structure is going to be solved in a reasonable time given current computing capabilities. Thus, virtually all software packages used in cryo-EM employ algorithms that accelerate the minimization process by avoiding exhaustive searches. As a consequence, it is possible for a program exploring the solution landscape to converge closer to a local minimum than to the “real” answer thus placing the molecular image in the wrong orientation. This phenomenon was particularly insidious before the use of lower noise direct electron detectors and before the introduction of “gold standard” methods for refinement. This has led to the publication of a number of incorrect structures in the resolution range of 10 to 20 Å.

The EMAN tutorial is a good source of information regarding image processing using Cryo-EM, but it should not be taken as the authoritative guide in the field. Even though “fake maps” may seem consistent with low resolution tertiary structural information, the degree of confidence of a map obtained by cryo-EM increases substantially once secondary structure can be resolved. At about 8 Å, the number of solved alpha-helices is very informative, especially if conformers or homologs are known (e.g. Matthies D, Dalmas O, **Borgnia MJ**, Dominik PK, Merk A, Rao P, Reddy BG, Islam S, Bartesaghi A, Perozo E, Subramaniam S. Cell. 2016 Feb 11;164(4):747-56.). A structure at 5 Å or better, characterized by the presence of discernible beta-strands, is strongly suggestive of a real biochemical structure. As shown in other portions of this rebuttal and in the manuscript itself, we were able to assign primary structure to a map derived from biased data but nevertheless isotropic. The authors find it difficult to imagine a situation in which a map arising from a “local minimum in energy space” is fully consistent with the secondary structure of homologous AAA ATPase structures and the primary structure derived from its biochemical composition.

We appreciate the suggestion of using a filtered structure of a homolog like P97 as the initial reference for 3D refinement. However, we should point out that proper filtration and high-resolution phase randomization of the P97 model, as necessary to prevent model bias, results in a low-resolution reference that closely resembles the one obtained by applying the “Initial Model” generation algorithm to our data. The authors believe that, under these circumstances, using the actual data is always the preferred option.

Point 2.3) *It is unclear to this reviewer why the authors decided to run their autopicking routines with a gaussian blob? Relion offers the possibility to generate templates by manually picking a few thousand of images. Based on Suppl. Fig 3, the 2D classification step throw away ~70% of the picked particles. I think that result is a clear sign that the autokicking step was not well optimized. Also, a gaussian blob certainly will not favor picking side-view particles. They are very different from a gaussian blob. Consequently, it is likely that the possible orientation bias existing with Rix7 was aggravated with the autokicking approach undertaken by these authors. Authors should at least explain the rational for her autopicking approach.*

The reviewer is correct that there are a number of different ways to pick particles through various software packages. We attempted a variety of particle picking algorithms to pick Rix7 particles. We went with the gaussian blob for particle picking because we found it gave us more distinct orientations of Rix7 particles. This was especially important for the low defocus micrographs. The failure to pick up side-views of Rix7 by using gaussian blob (see the example below) was not an issue. Following the reviewer’s suggestion, we also generated a template by manually picking ~1000 particles, followed by autopicking with a template using CryoSPARC. From this approach 270,460 putative particles were picked in the initial round. We followed the same refinement strategy used for the gaussian blob picked particles (supplementary Figure 3) and we were able to generate a final map with a similar quality and resolution as before. Shown below are two representative micrographs from our Rix7 dataset with different autopicking approaches (gaussian blob in Relion and template in CryoSPARC).

DF = -1.25 μm

RELION Gaussian Blob

CryoSPARC Template

DF = -2.46 μm

RELION Gaussian Blob

CryoSPARC Template

Point 2.4) *In the class2D steps: How many classes were requested for the first step of classification before the dataset was split into top and side views? Did they only request 10-15 classes as in the second step? Why so few? For a dataset of this size with over half a million particles a much larger number of classes is typically used (~100-150).*

In the first round of 2D classification we requested 50 classes for the down-scaled particles (pixel size binned by 4). The number of classes needed depends on both the resolution and size of the particle. We down-scaled the particles to speed up computation. In this case, requesting 50 classes was sufficient to generate top, bottom, and side views of Rix7. We also observed a number of classes that have a black background, indicating that all the particles collapse into fewer than 50 classes. For the final step of 2D classification we classified the top/bottom and side views separately. We requested 120 classes for the top/bottom view and 10 classes for the side views. Only the selected 2D views (boxed in red) are shown in Supplemental Figure 3b.

Shown below is the 2D classification for the side view:

10 classes for the side view

Shown below is the 2D classification for the top/bottom view:

120 classes for the top/bottom view

Red boxes indicate particles that were used for 3D refinement.

Point 2.5) *A limitation of the obtained structure was that the NTD of Rix7 protomer was not visible, presumably due to flexibility. Did the authors attempted focus classification with signal background subtraction in these regions to attempt describe these structural motifs?*

As mentioned above under the response for Reviewer #1 (Point 1.1), the NTD of Rix7 contains ~100 residues that are predicted to be disordered. These residues lie b/w the very N-terminal 3-helix motif for which there is an NMR structure and the D1 AAA domain. In the absence of a stabilizing co-factor we do not expect to see density for the NTD. Moreover, we do not observe any weak or fragmented density above the top of the D1 domain which may correspond to the NTD. This limits the effectiveness of focused refinement as there is nothing to focus on. Despite this we still attempted focused 3D classification using protomers B-D, which are the most well-ordered in the reconstruction, as the reference mask without image alignment, but we still did not see any additional density for the NTD.

Point 2.6) *Regarding the density in the axial channel presumably representing a client protein being pull out and thread it through the pore is an interesting finding. However, if I understood correctly the description from the authors, a substrate would be required to always be threaded in the same way through the pore in order to produce a define density. Are the authors proposing that as the substrate is treaded is handed over from the pore loop in one protomer to the next in a specific sequential manner? Are they suggesting that because of the asymmetric nature of the hexamers threaded substrate always enter and progress through the axial pore in a define sequence through specific protromers? This would be as opposed to what is proposed in the manuscript that there are 6 equivalent positions in the ring that the substrate could use as it is translocated through. In this last case, cryo-EM would not provide a defined density but rather a fragmented density resulting from the averaging of particles with substrates bound on all six positions. The authors should elaborate about this point in their discussion.*

The reviewer is absolutely correct that there are 6 equivalent positions within the ring that the substrate could use as it is translocated. We apologize if our discussion was confusing and we have re-written this section of the discussion to make this point clear (Please see Page 14, first paragraph). As is the case for several other recently published AAA-ATPases with substrates bound, we believe that the density is ambiguous from averaging the particles with substrates bound on all six positions.

Reviewer #3 (Remarks to the Author):

What are the major claims of the paper?

The manuscript by Lo et al., "Unraveling the Mechanism of Substrate Processing by the AAA-ATPase Rix7" describes in details the structure of the AAA+ C.thermophilum Rix7, an ATPase involved in the production/maturation of the large 60S ribosomal subunit by removal of the assembly factor Nsa1. The authors present the structure of the hexameric Rix7 assembly in complex with a threaded substrate. They therefore determine that the protein is definitively an unfoldase. They deduct the unfolding mechanism from the structure and describe the specific and different roles of the two AAA domains.

Are the claims novel? If not, please identify the major papers that compromise novelty

The manuscripts contains novel results but the major claims are actually not so novel. Here below I explain more in details my point of view hoping to help the authors to bring forward the more novel findings contained in the manuscript.

The structure of Rix7 is new as, apart from the N-terminal NMR structure of NVL2 (Fujiwara et al, JBC 2011) and the AAA-2 of NVL2 (pdb code 2x8a (3 helices Nter) no other structural

information about Rix7/NVL2 was known. However, the oligomeric arrangement of Rix7 is almost identical to that of other AAA+ unfoldases recently studied by EM. A number of two-cassette AAA+ unfoldases structures have been solved in the last years and they all show the same unfolding mechanism with one “seam” subunit of the AAA+ rings that perform the pulling force. All these structures are also all analysed in a similar way with particular emphasis to the pore substrate-binding loops organisation (Fig 4a-c in this manuscript) and the relative position of the 6 subunits within the hexameric ring (Fig 3 in this manuscript). Even though it is correct and thorough to preform all the analysis of the canonical AAA+ features to find whether or not the Rix7 looks similar to the already known AAA+ unfoldases, once found that the structural features of subunits organisation, pore loops and nucleotides pockets are actually very similar to all other AAA+ unfoldase structure solved so far, the authors should have acknowledge it more and stress instead more the discovery of structural features specific of Rix7 and indeed novel and interesting. The authors analyse two specific structural features of Rix7, the well-conserved serine in the PLI loop of AAA1 and the post alpha7 insertion, that should be given more emphasis in the text and figures as they would strengthen the manuscript and make it more interesting for researcher interested in the specific biological function of Rix7, ribosome large subunit maturation. AAA proteins have very well conserved motors, but perform extremely different biological activities, so a lot of novelty is in the details that differentiate one motor from the other.

Will the paper be of interest to others in the field?

The paper will be of interest to others in the field of structural biology specifically in the field of ribosome maturation, chaperones in general and of course AAA proteins. The paper could also be of interest from an evolution point of view as it adds information about a specific class within the large AAA+ superfamily.

Will the paper influence thinking in the field?

This is hard to tell. Might affect the thinking in terms of understanding the steps of ribosome formation.

Are the claims convincing? If not, what further evidence is needed?

The claims are convincing, they are strongly based on previous literature and find nice corroboration in the structural visualisation. The combination of functional in vivo experiments with the structural analysis is well performed.

Are there other experiments that would strengthen the paper further? How much would they improve it, and how difficult are they likely to be?

Some analysis of the N-terminus of Rix7 would be appreciated. More specific comments are found below. Are the claims appropriately discussed in the context of previous literature?

The authors cite the existing literature and analyse the structure of hexameric Rix7 in a fairly canonical way complying with all the requirements so far established in the field (analysis of AAA defining motifs such as walker A, B, R-finger, nucleotide binding , substrate binding). They refer to the literature correctly, but they tend to tune down the similarities of their structure to all the deposited and published maps if similar unfoldases. The authors should not worry about being completely honest about the extreme similarity of Rix7 with other AAA+ structure as Rix7 is a different protein, performs a different activity and in fact the paper also contains interesting and novel information about the specific structural features of Rix7.

If the manuscript is unacceptable in its present form, does the study seem sufficiently promising that the authors should be encouraged to consider a resubmission in the future?

The manuscript is already in a form acceptable for publication, but changes (details below) might improve it.

Is the manuscript clearly written? If not, how could it be made more accessible?

The manuscript is clearly written.

Could the manuscript be shortened to aid communication of the most important findings?

The manuscript is already in a fairly compact form.

Have the authors done themselves justice without overselling their claims?

The authors have done themselves justice, sometimes up-tuning and sometimes down-tuning the novelty of some results.

Have they been fair in their treatment of previous literature?

Yes, they have.

Have they provided sufficient methodological detail that the experiments could be reproduced?

For the cryo-EM parts, I think they did. Some choices should be justified (see details below).

Should the authors be asked to provide further data or methodological information to help others replicate their work? (Such data might include source code for modelling studies, detailed protocols or mathematical derivations).

The authors should have already deposited the map and the pdb models and the EMDB and PDB codes should be in stated in the manuscript.

Comments along the text and figures:

Point 3.1) *In the introduction the authors limit their structural comparison to NSF, p97 and Pex1/Pex6. This, even though formally correct as these proteins are the closest relative to Rix7 within the AAA+ superfamily, it is a bit misleading as there are nowadays various cryo-EM maps of two-cassette AAA+ proteins. The article refers to these works later on, but in the introduction only NSF, p97 and Pex1/Pex6 are mentioned. It appears as a way a bit forced of adding novelty to the findings. The structure of VAT, which belongs to the same class and has a substrate-bound must also be mentioned in the introduction for completion. Likewise the introduction states “The structure reveals an unexpected asymmetric configuration ...”. Asymmetric configurations of AAA+ rings are not unexpected anymore. Also the fact that 5 out of 6 subunits grab the substrate is not unexpected. The authors should tune down the novelty of the structural arrangement of Rix7 in the last paragraph of the introduction. Tuning it down should not undermine the novelty and beauty of the Rix7 structure anyway because it is true that it is for the first time Rix7 is visualised and clearly shown to be a unfoldase. The authors should include at the end of the introduction the results about the specific features of Rix7, the are in the discussion. I think that these results are novel and add knowledge to the process of understanding how the same AAA molecular motor evolved to perform such different activities. This is one of the main open questions in the AAA field and the author could stress their contribution to it in the manuscript.*

We thank the reviewer for their support of our manuscript and for the wonderful suggestions on shifting the emphasis of the results/discussion. Following the suggestion of the reviewer we have modified the text in several places to emphasize what is novel about the Rix7 structure and we include more comparisons of Rix7 to other type II AAA-ATPase proteins (Please see the abstract, final paragraph of the introduction, and discussion). We have expanded the

introduction to include a more comprehensive introduction to type II AAA-ATPase proteins including the ancestral VAT.

Point 3.2) Paragraph "ATPase activity of Rix7 modulates 60S subunit formation". Elegant experimental setup with the regulable Rix7 endogenous expression for the analysis of the various mutants *in vivo*. It is not clear how many times were the sucrose gradient experiments repeated, should be said.

Each sucrose gradient was repeated at least three times, this has now been indicated in the methods (Pages 17, last paragraph).

Point 3.3) Paragraph "Asymmetric hexameric architecture of Rix7". For the model building, did the authors check the structure of the AAA2 domain of NVL2 (pdb code 2x8a)? It is not published but the coordinates are deposited. How similar is it to the manuscript model?

We were aware of this PDB but neglected to mention it in our initial submission. The overall AAA fold is similar as one would expect but the two subdomains are in different orientations. This comparison is now mentioned in the text (page 8, first paragraph and page 11, last paragraph) and shown in Supplemental Figure 5c.

Point 3.4) The N-terminus of Rix7 is not visible at all. Did the authors try to analyze via focused classification the terminal region of the molecule?

As mentioned above for both Reviewer #1 and #2 (Please see Point 1.1 and 2.5), the NTD of Rix7 contains ~100 residues that are predicted to be disordered. These residues lie b/w the very N-terminal 3-helix motif for which there is an NMR structure and the D1 AAA domain. In the absence of a stabilizing co-factor we do not expect to see density for the NTD. However, following the suggestion of reviewer #2 and #3 we did try focused classification around the top of the D1 domain, but we could not see any additional density for the NTD.

Point 3.5) The sentence "The individual AAA domains all superimpose well with one another but there are differences in the relative orientations between D1 and D2 AAA domains, which gives rise to the spiral arrangement of the protomers" is unclear. From this sentence it seems as if the small and large subdomains of each individual AAA superimpose well which is not the case as later explained in Figure 5 and SuppFig 5b. Moreover, difference in the relative orientation of D1 and D2 give rise to two different types of spiral arrangements in the D1 and D2 ring, not to the spiral as such. One could have difference in the relative orientation of D1 and D2 and have a planar ring.

We apologize that this description was confusing and have reworded the text to make this point more clear (Please see Page 7, last paragraph).

Point 3.6) Paragraph "Rix7 threads substrates through its central pore". The fact that Rix7 contains a substrate of unknown origin indicates that it is fairly non-specific as unfoldase. Can the author comment on it? The paragraph that describes the particularity and substrate contacts of the pore loops of Rix7 even when the aromatic residue in the loop triad for substrate binding is lacking, is interesting and novel and should be put more in light. The authors could make a dedicated panel showing the densities and the contact in between h-S-G and the substrate. This novel result should be also spelt out in the introduction and abstract in my opinion.

We thank the reviewer for the suggestion to highlight the novelty of the D1 pore loop. This is now mentioned in the abstract and the introduction (page 4, last paragraph). We agree that it is a significant finding that the D1 domain of Rix7 threads substrates in the absence of an aromatic residue. We have expanded the view for Figure 4c to include the interaction between the polypeptide and the D1 pore loops with the density. Moreover we now include an additional point mutant (S280A) within this motif, which further highlights the significance of the D1 pore loop signature motif.

It is hard to speculate on the specificity of the Rix7 unfoldase. The reviewer is correct that because we trapped Rix7 with a substrate this would suggest it lacks specificity. However we trapped a substrate using a mutant of Rix7 and while working at fairly high (~0.5 mg/mL) protein concentrations. Reconstitution studies will be needed in the future to fully address the question of substrate specificity.

Point 3.7) Paragraph “Rix7 pore loops are essential in *S. cerevisiae*. The hypothesis that the pore loops of Rix7 play a role in substrate selection is interesting. It is a novel point of the paper. Is it possible to run in vitro substrate affinity tests using putative substrates? Rix7 clearly binds to some substrate here during purification. The density is not good enough to determine the aa sequence of the substrate bound, but did the authors try to run mass spectrometry experiments to attempt determining the identity of the bound substrate? Do the author think that an adaptor protein is also playing a role in vivo?”

We would absolutely love to know the identity of the mystery substrate trapped by Rix7! Upon observing density in the central channel, MALDI-TOF mass-spec was the very first thing we tried to determine the identity of the substrate. The most abundant peptide that we could detect corresponded to the very C-terminus of Rix7. Mass-spec did not reveal any other peptides, aside from those arising from Rix7 in significant abundance. We think most likely the peptide in the channel represents Rix7 unfolding itself as was observed in the recent structure of VAT but this is difficult to prove conclusively.

Substrate affinity tests are ongoing in the lab but are currently a huge technical challenge. The number one problem hindering this is that WT-Rix7 is not very stable and difficult to purify in high enough quantities to carry-out unfoldase and ATP hydrolysis assays. Furthermore we and others have found that recombinant SC Rix7 and SC Nsa1 do not associate with one another in vitro (Lo *et al* Structure, 2017 and Kressler *et al* JCB, 2008). This strongly suggests that the interaction between these two proteins is mediated by either a co-factor or post-translational modification. Cdc48/p97 is dependent upon a post-translational modification (ubiquitin) and co-factor binding to recognize its numerous targets. We therefore hypothesize that Rix7 employs additional mechanisms to ensure correct substrate selection in vivo.

Point 3.8) Paragraph “The seam promoter and the nucleotide-bound state”. Is there enough resolution to claim that the walker A P loop adopts different orientations? This is difficult to judge without having the map. The authors should show a figure with the density map for this area. E.g. Figure 5 c should show the actual densities.

We thank the reviewer for this suggestion and this figure has been updated to include the density. The density for the Walker A P-loop backbone is well defined in P4 in P5 and its clear that the P-Loops in these protomers are in different conformations (apo vs ATP bound). The density for P6/seam is less well defined, the seam protomer has a lower local resolution and the ATP is not well ordered. Therefore we removed the P6 protomer from the superposition in Figure 5C and 5D.

Point 3.9) Paragraph “Insertion following alpha7 in the D1 and D2 domains”. The analysis of the insertion after alpha 7 is interesting and novel. The authors should in my opinion stress it in the introduction/abstract and make SuppFig 6 (or a variant of it) a main figure. Are there some mutations that the authors could make on the post alpha 7 insertion to somehow “block” the N-terminus (for cryo-EM studies)?

We thank the reviewer for the suggestion to highlight more of the significance of the post alpha 7 insertion. We have moved this to Figure 6 in the main text and highlight this finding in the abstract and the final paragraph of the introduction. We also made truncations to the post-alpha 7 insertion in both the D1 and D2 domains for yeast complementation assays (Figure 6c and 6f), which revealed that both insertions are important for Rix7 function *in vivo*.

We don't know if the post alpha 7 insertion is involved in positioning of the NTD of Rix7. All we can say from our structure is that within the D1 domain this insertion clashes with the position of the NTD of p97/Cdc48. Because there is no similarity between the Rix7 and p97/Cdc48 NTDs it is hard to speculate on the potential location of the Rix7 NTD. To visualize the NTD of Rix7 in the future we will likely need to first identify stabilizing co-factors.

Point 3.10) Of all the 3D classes obtained none, not even at low resolution and threshold included the N-terminus?

Unfortunately no, despite many different attempts we were never able to observe the N-terminal domain (Please see Point 1,1 and 2.5 for more details).

Point 3.11) Discussion. The claim that Rix7 shows a unique combination of substrate threading by D1 and processing translocation by D2 is not sufficiently funded by the structure nor explained with words. Should be clarified.

We agree with the reviewer that this sentence was confusing and not supported by our structure, therefore this sentence was removed from the discussion.

Point 3.12) Did the authors try to perform 3D classification as first step, even before 2D?

No, we did not attempt to perform 3D classification as the first step before 2D because it was needed to produce a high quality scattering map of Rix7.

Point 3.13) They state they refined the model in reciprocal space. Can they confirm they did not touch the map? The map and pdb must be deposited.

We refined in reciprocal space so that we could utilize the CNS software for refinement. We absolutely did not touch the map. Both the map and PDB have been deposited and the accession codes (PDBID 6MAT and EMD-9063) are included in the revised manuscript (Please see Table 1).

Point 3.14) More general questions. In a previous work the authors describe interaction of WDR74 and NVL2 via the AAA1 ring. Looking at the Rix7 structure can they make any comment/rational about the nature of this interaction?

Based upon our earlier work, we assume that WDR74 binds to the top of the NVL2-AAA1. We were able to show previously WDR74 binds to the alpha/beta subdomain of the D1 domain of

NVL2. Based upon our cryo-EM structure, the alpha/beta subdomain would only be accessible from the top of the D1 domain. This position at the top of the ring would ideally position WDR74 for unfolding in the central channel.

Point 3.15) *From the structure and the biochemical experiments here performed any idea on whether Nsa1 is a cofactors rather than a substrate?*

Our working hypothesis is that Nsa1 is a substrate and not a co-factor. This is based on the following information: (1) The association between the mammalian homologues of Nsa1 and Rix7 is nucleotide dependent and (2) Mutation of Rix7 traps Nsa1 on pre-60S particles. Hopefully in the future we will be able to use an *in vitro* reconstitution system to fully address this question.

Figures:

Point 3.16) *Figure 1 and 2 ok Figure 3 ok , but it would be nice to show an insight with the density for the nucleotide (if/when visible)*

No changes were made to Figures 1, 2, and 3.

Point 3.17) *Figure 4 a) this panel does not add more information than the panel b as it is generated anyway from the model. Might be nicer to show the actual density from the EM map.*

We chose to leave Figure 4A in the revised manuscript because we felt it is important to illustrate that the 4 pore loops of the P6 protomer do not contact the substrate. The actual density for the polypeptide is shown in Figure 3c. We used the difference map for Figure 4a because we wanted to illustrate that after fitting in the D1 and D2 domains for all 6 protomers there is still significant density in the central channel.

Based on the suggestion of the reviewer we modified Figure 4c, to show the details of the interaction between the D1 PL1 and the substrate with the density overlaid.

Point 3.18) *Figure 5 c should be changed and real density of the walker A areas should be shown rather than the model.*

As mentioned above, under Point 3.8 we have modified this figure to include the density for the P-Loop from P4 and P5 protomers.

Point 3.19) *Supp Fig1 and 2 ok*

Supplementary Figure 2 now includes the western blots for the additional Rix7 mutants. Uncropped blots are shown in Supplementary Figure 6.

Point 3.20) *Supp Fig3 Why was C2 symmetry applied in the first place?*

Prior to solving the structure of Rix7 we did not know the symmetry of the particle but suspected that it would be C6 based on the similarity to p97/Cdc48/VAT. We used C2 symmetry at the beginning because it allowed us to get a better signal and this was useful for centering and re-extraction of the particles. After this point we relaxed the symmetry and discovered that Rix7 is asymmetric.

Point 3.21) *Supp Fig4 the FSC of masked, unmasked, phase randomised maps should be shown. The angular distribution still shows some preferential orientation Did the authors do anything to balance the views during image processing? If so, they should describe.*

Aside from separating the top/bottom and side views during 2D classification (please see Supplementary Figure 3) we did not do anything else to balance the views during image processing. We have updated Supplemental Figure 4 to include the FSC of the masked, unmasked and phase randomized maps as well.

Point 3.22) *Supp Fig5 ok*

Supplemental Figure 5 has been modified to include a superposition with the NVL2 D2 deposited crystal structure.

Point 3.23) *Supp Fig6 with changes should become a main figure.*

We thank the reviewer for the suggestion and Supplemental Figure 6 is now Figure 6 in the main text. This figure has also been updated to include the growth curves for the post alpha 7 deletions.

REVIEWERS' COMMENTS:

Reviewer #1 (Remarks to the Author):

Lanes 35 and 91/92: There is nothing like "canonical unfoldase residues". This has to be changed to for example "residues conserved in unfoldases" or similar

Lane 186: Is "However" required here?

Lane 188: Why not state the same as mentioned in response to reviewer 3: i.e. that the most abundant peptide detected by MS originates from the C-terminal end of Rix7 itself? This is an important information.

Lane 188: delete "likely". Since the protein was isolated after heterologous expression in bacteria it is sure that it cannot be the substrate!

Lane 309/310 and 319: "remains a mystery" is not a good formulation for a scientific journal and should be changed

Lanes 423/424: "Starter cultures grown in presence of DOX(120µg/ml)". I am sure this is a mistake and the authors added DOX only to the main culture and not already to the starter cultures.

Reviewer #2 (Remarks to the Author):

The authors have addressed the concerns of this reviewer. I think the paper should be accepted now for publication as is.

Thank you for the opportunity to review this beautiful manuscript.

Reviewers' comments (*italics*) and our reply (**red**).

Reviewer #1 (Remarks to the Author):

We thank the reviewer for their careful review of our revised manuscript and we have made all of their suggested changes to the manuscript.

Lanes 35 and 91/92: There is nothing like “canonical unfoldase residues”. This has to be changed to for example “residues conserved in unfoldases” or similar

We thank the reviewer for this suggestion and have replaced the phrase “canonical unfoldase resdiues” with aromatic-hydrophobic motif.

Abstract: The structure establishes that type II AAA-ATPases lacking the aromatic-hydrophobic motif within the first AAA domain can engage a substrate throughout the entire central channel.

Lanes 91/92 now reads as follows: Despite the absence of the aromatic-hydrophobic motif found in pore loops of unfoldases within the D1 domain, the Rix7 pore loops from both the D1 and D2 domains grip the polypeptide through distinct motifs which are essential for Rix7 function in vivo.

Lane 186: Is “However” required here?

As suggested by the reviewer we have removed the word however.

Lane 188: Why not state the same as mentioned in response to reviewer 3: i.e. that the most abundant peptide detected by MS originates from the C-terminal end of Rix7 itself? This is an important information.

We now include this information in the text. We agree with the reviewer that this information is important.

Lane 188: delete “likely”. Since the protein was isolated after heterologous expression in bacteria it is sure that it cannot be the substrate!

We have removed the term “likely”

Lane 309/310 and 319: “remains a mystery” is not a good formulation for a scientific journal and should be changed

We have removed the phrase “remains a mystery” and simply state that it is unclear.

Lanes 423/424: “Starter cultures grown in presence of DOX(120µg/ml)”. I am sure this is a mistake and the authors added DOX only to the main culture and not already to the starter

cultures.

We agree with the reviewer that it seems unusual to add DOX to the starter culture, however it was necessary in this case. Rix7 is an essential gene, which means we cannot knock it out, we can only knock it down. We found that pre-treatment with DOX overnight was a requirement to observe reproducible effects by sucrose gradient fractionation. To ensure that DOX does not have a negative effect on cells we included the WT-Rix7 control for both our growth assays and sucrose gradients. As you can see in Figures 1 and 2. The WT Rix7 plasmid grows in both the presence or absence of DOX and we observe a normal polysome profile. Similar approaches have been used to characterize other essential ribosome assembly factors (Belhabich-Baumas, et al NAR 2017; Soudet, et al EMBO 2010; Castle, et al NAR 2013)

Reviewer #2 (Remarks to the Author):

The authors have addressed the concerns of this reviewer. I think the paper should be accepted now for publication as is.

Thank you for the opportunity to review this beautiful manuscript.

We thank the reviewer for their support of our manuscript!